



# Uncertainty in predicting the Eurasian snow: Intercomparison of land surface models coupled to a regional climate model

Da-Eun Kim[1,3,4] and Seon Ki Park[1,2,3,4]

[1]Department of Climate and Energy Systems Engineering, Ewha Womans University, Seoul, Republic of Korea
[2]Department of Environmental Science and Engineering, Ewha Womans University, Seoul, Republic of Korea
[3]Severe Storm Research Center, Ewha Womans University, Seoul, Republic of Korea
[4]Center for Climate/Environment Change Prediction Research, Ewha Womans University, Seoul, Republic of Korea

**Correspondence:** Seon Ki Park (spark@ewha.ac.kr)

**Abstract.** Variability of large and synoptic scale circulations in Asia is strongly affected by the winter and spring Eurasian snow. Therefore, an accurate prediction of the Eurasian snow is of the utmost importance in predicting the climate and weather phenomena in Asia. Most global/regional models are coupled with several land surface models (LSMs) in which the land surface process parameters are calculated under their own physical principles and parameterization schemes. In this study, using the Weather Research and Forecasting (WRF) model, we make intercomparision of LSMs in terms of simulating the Eurasian snow. Simulations are carried out from 1 June 2009 to 31 August 2010, including a spin-up time of 6 months, by employing four different LSMs — the Unified Noah LSM, the Noah LSM with multiparameterization options (Noah-MP), the Rapid Update Cycle (RUC) LSM, and the Community Land Model version 4 (CLM4). The NCEP Final (FNL) Operational Global Analysis data are used as initial and boundary conditions. The LSM results are evaluated using the Canadian Meteorological Centre Daily Snow Depth Analysis Data, the Moderate Resolution Imaging Spectroradiometer (MODIS)/Terra Snow Cover Monthly L3 Global 0.05Deg Climte Modeling Grid (CMG) Version 6, and the MODIS Bidirectional Reflectance Distribution Function (BRDF)/Albedo Product. Although all the LSMs represent reasonable results, the Noah-MP represents the most accurate predictions in all three variables (snow depth, fractional snow cover, and albedo), in terms of not only quantitative aspects but also spatial correlation patterns. Our results indicate that prediction of the Eurasian snow cover is sensitive to the choice of LSMs coupled to the global/regional climate models, and hence the future climate projections.

## 1 Introduction

The snow cover state over the land surface has an appreciable impact on the large and synoptic scale hydrological cycles and surface energy budgets (Dewey, 1997; Cohen and Rind, 1991; Cohen, 1994). As snow cover on the land surface has characteristics of high albedo and thermal emissivity, small roughness length, and low thermal conductivity, it consequently has a feedback mechanism with changes in weather and climate (Wu and Qian, 2003). Snow inhibits direct heat exchange between land surface and atmosphere: in winter the land surface temperature is higher than the air temperature, whereas in spring the former becomes lower than the latter since snow reflects solar radiation with its high albedo (e.g., Park et al., 2017). The temperature gradient between the land surface and the ocean is one of the important factors that influences the atmospheric cir-





culations. Therefore, snow over the land surface during winter and spring plays an essential role in regional/global atmospheric circulations through variation of the surface air temperature. Furthermore, the snow melting process significantly influences the regional water and energy cycle (e.g., Cassardo et al., 2018a, b). Melted snow can be an important source of the water resources; however, it can also result in hazards such as soil erosion and flooding.

Snow in the Northern Hemisphere (NH) is mostly concentrated on Eurasia and North America. The Eurasian snow, which occupies a large portion in NH (Chen et al., 2016), is highly related with the global climate system through interactions with the atmospheric fields. Climatic aspects, associated with the El Niño-Southern Oscillation (ENSO) and the Arctic Oscillation (AO), have been studied for their strong relationship with the Eurasian snow (Barnett et al., 1998; Furtado et al., 2015). Furthermore, the Eurasian snow is highly related with the climate and weather systems in Asia. Many studies have been conducted on the

correlation between the South Asian climate/weather systems and the Eurasian snow. The Indian summer monsoon rainfall is one of the most well known phenomenon which highly interacts with the Eurasian snow (e.g., Hahn and Shukla, 1976; Dey and Bhanu Kumar, 1982; Ropelewski et al., 1984; Dey et al., 1985; Khandekar, 1991; Parthasarathy and Yang, 1995; Vernekar et al., 1995; Sankar-Rao et al., 1996; Yang, 1996; Bamzai and Shukla, 1999; Kripalani and Kulkarni, 1999; Liu and Yanai, 2002; Wu and Qian, 2003; Fasullo, 2004; Peings and Douville, 2010). The interactions between the East Asian climate/weather

systems and the Eurasian snow also have been studied (e.g., Yang and Xu, 1994; Kripalani et al., 2002; Wu and Qian, 2003; Zhang et al., 2004; Zhao et al., 2007; Won et al., 2017). Therefore, more accurate prediction of the Eurasian snow will improve the forecast of weather and climate systems over Asia.

The land surface plays a vital role in the interactions between atmosphere and various land surface processes, including soil and vegetations, whose parameters are mostly calculated by the land surface models (LSMs). Most regional/global models

have several LSMs as optional choices. The coupled LSM provides the bottom boundary conditions for the atmospheric models — e.g., surface sensible heat flux, surface latent heat flux, upward longwave radiation, and upward shortwave radiation. The LSMs include many subgrid-scale physical processes that require parameterizations, and have their own characteristic physical processes with unique land surface/soil/vegetation parameters. Therefore, different choice of LSM brings about different results in the land surface variables/parameters, influencing the related atmospheric variables, even with the same initial and boundary

conditions (e.g., Henderson-Sellers et al. , 1993; Chen et al., 1996; Rodriguez-Camino and Avissar, 1998; Bastidas et al., 2006; Trier et al., 2008; Jin et al., 2010; Jimenez et al., 2011). Moreover, the performance of each LSM varies depending on the terrain and land cover/use over the domain to be simulated (e.g., Oleson et al., 1997; Kato et al., 2007). Each LSM has its own complexity of formulation in the snow scheme. In this study, we evaluate the LSMs coupled to a regional climate model in predicting the Eurasian snow, especially focusing on snow depth, fractional snow cover, and surface albedo. The followings

are the goals of this study: (1) to understand the structure of snow scheme in each LSM; (2) to analyze why different LSM produces different results in snow-related parameters; (3) to evaluate the performance of each LSM to predict the Eurasian snow; and (4) to suggest some approaches to further improve the model performance.





## 2 Method and data description

### 2.1 Model description

For the regional climate simulations, we employ the Weather Research and Forecasting (WRF) model coupled with four different LSMs to simulate the Eurasian snow during the period from 1 December 2009 to 31 May 2010. A summary of the WRF model configuration, including the LSMs, is provided in Table 1.

#### 2.1.1 WRF model configuration

All the simulations are performed by the Advanced Research WRF (WRF-ARW; Skamarock et al., 2008) version 3.6.1 with a horizontal spacing of 30 km and a time step of 180 minutes. The computational domain contains the Eurasia region by the Lambert conformal projection with 289 grid points in the east-west direction and 184 grid points in the south-north direction (see Fig. 1a). The model atmosphere is discretized with 30 vertical eta levels with thinner layers at the bottom and thicker layers at the top, and the model top is set at 50 hPa. The number of soil layers is different for a choice of different LSM. The experiments are carried out with the WRF single-moment 3-class microphysics scheme (WSM3; Hong et al., 2006) — a simple phase scheme that includes prognostic water substance variables (i.e., water vapor, cloud water/ice, and rain/snow). The radiation fluxes are calculated by the Dudhia shortwave radiation scheme (Dudhia, 1989) and the Rapid Radiative Transfer Model (RRTM) longwave radiation scheme (Mlawer et al., 1997). For the planetary boundary layer (PBL), we apply the updated Yonsei University (YSU) PBL scheme (Hong et al., 2006; Hong and Kim, 2008) which considers the nonlocal fluxes implicitly through a parameterized nonlocal term. For the convective parameterization, we employ the Kain-Fritsch (KF) scheme (Kain and Fritsch, 1993; Kain, 2004) based on the Lagrangian parcel method.

#### 2.1.2 The Unified Noah land-surface model

The Noah LSM has been developed from 1990s through a collaboration of many private and public investigators. The collaboration included the Office of Hydrological Development of the National Weather Service, the National Environmental Satellite Data and Information Service, the National Aeronautics and Space Administration (NASA), the National Center for Atmospheric Research (NCAR), the U.S. Air Force, and the Oregon State University and other university partners spearheaded by the National Centers for Environmental Prediction (NCEP) (Chen et al., 1996; Koren et al., 1999; Chen and Dudhia, 2001a, b; Ek et al., 2003).

The Noah LSM consists of 4 soil layers and 1 snow layer lumped with the top soil layer. To predict snow variables, the Noah LSM starts from new precipitation. Based on the forcings from the atmospheric model, the Noah LSM determines the type of precipitation into three types — rain, freezing rain, and snow. Through this process, once the Noah LSM detects new snowfall, the model changes snowfall amount to liquid equivalent snow depth. The snow parameterization is based on the energy and mass balance of snow pack (Chen et al., 1996; Koren et al., 1999):

$$\frac{dW_s}{dt} = P_s - M_s - E \tag{1}$$



$$M_s = \frac{1}{L}(Q_{sw} + Q_{lw} - Q_{lt} - Q_{sn} - Q_g) \tag{2}$$

where $W_s$ is snow water equivalent, $P_s$ is precipitation in the form of snow, $M_s$ is snowmelt rate, $E$ is snow evaporation, $Q_{sw}$ is net solar radiation, $Q_{lw}$ is net longwave radiation, $Q_{lt}$ is latent heat flux, $Q_{sn}$ is sensible heat flux, $Q_g$ is ground heat flux, and $L$ is latent heat of fusion. However, with this original snow parameterization, snow depth and snow area were overestimated,

causing biases in the snow-soil surface heat exchange. In order to overcome this problem, Koren et al. (1999) introduced the compaction parameterization that considers the snow density change due to compaction.

Snow melting and new snowfall also have to be considered in calculating snow density. When the snow surface temperature becomes higher than 273.15 K, the Noah LSM updates snow density by considering the snow melting process with an assumption that a 13% of liquid water per day can be stored in snow during snow melt (Koren et al., 1999). The density of new

snowfall is calculated by an equation, dependent on air temperature, from Gotleib (1980). Then, snow depth is calculated from snow water equivalent and snow density; fractional snow cover is calculated from snow water equivalent and tuning parameter. Surface albedo, including the snow effect, is calculated by using fractional snow cover, snow albedo and snow-free albedo. The wavelength-integrated albedo represents the highest value immediately after new snowfall, and it tends to decline as time passes without snowfall — see Livneh et al. (2010) for an albedo decaying scheme.

### 2.1.3 The Noah-MP

The Noah LSM with multiparameterization options (Noah-MP) provides multiple options for the land-atmosphere physical interaction processes. Compared to the Noah LSM, the Noah-MP especially improved in reproducing surface fluxes, skin temperature over dry periods, snow water equivalent, snow depth, and runoff (Niu et al., 2011).

Similar to the Noah LSM, the Noah-MP is also based on the energy and mass balance of snow pack in the snow accumula-

tion/ablation parameterization — see Eqs. (1) and (2). The Noah-MP contains 4 soil layers and 3 snow layers; thus, the snow scheme tends to be more complicated than the Noah LSM. When there is no snow-accumulated layer, it starts with the new snowfall. The Noah-MP partitions precipitation into rain and snow (Jordan, 1991), using the surface temperature and freezing/melting point that is set to 273.16 K in the model. In addition, the Noah-MP considers snow interception by the canopy since the capacity of capturing snowfall is much higher than that of capturing rainfall. By considering snow interception, the

model calculates snowfall at the ground which is the sum of the drip (unloading) rate for intercepted snow and the throughfall of snow. In calculating the throughfall, horizontal wind speeds and greeness vegetation fraction are also considered. Then, the Noah-MP calculates snow depth and snow water equivalent from snowfall at the ground. The bulk density of snowfall is calculated from the fresh snow density as a function of air temperature (Hedstrom and Pomeroy, 1998). When snow depth gets higher than 0.025m, the model starts to create a new snow layer and considers the compaction process which includes

destructive metamorphism, overburden, and melting effects.

The Noah-MP keeps updating snow layer ice and snow layer liquid water resulting from sublimation/evaporation, which are used in the compaction parameterizations. The ground snow cover fraction is then parameterized as a function of snow depth, ground roughness length, snow density and melting factor which is set to 2.5 (Niu and Yang, 2007). Lastly, sur-



face albedo, including the snow effect, is calculated through fractional snow cover, snow albedo and bare soil albedo by considering snow age, snow albedo, ground albedo, solar fluxes for the unit incoming direct and diffuse fluxes, and reflectance/transmittance/absorption by vegetation. Especially for snow albedo, the Noah-MP has two options — CLASS (Verseghy, 1991) and BATS (Yang et al., 1997). The CLASS updates snow albedo during new snowfall assuming that a 1 cm snow depth

will fully cover the old snow while the BATS updates it by considering snow age and calculating snow albedo for direct and diffuse radiation separately. For snow age, the Noah-MP considers three effects — the grain growth effect due to vapor diffusion, the grain growth effect at freezing of melt water, and the soot effect. Recently, Park and Park (2016) developed a new parameterization to improve the winter surface albedo by considering the effect of vegetation structure.

### 2.1.4 The RUC LSM

The Rapid Update Cycle (RUC) model was originally developed by NCEP as a mesoscale atmospheric data analysis and prediction system (Benjamin et al., 2004). In this version of WRF, the RUC LSM contains 6 soil layers and 2 snow layers with vegetation (Smirnova et al., 2000). It is also based on the original Eta model in the snow accumulation/ablation parameterization (Eqs. 1 and 2).

The RUC LSM starts from classification of precipitation type through the precipitation information from the microphysics
scheme. When the RUC LSM detects snow, precipitation is presented as a function of the snow ratio. On the other hand, when no snow is detected from the atmospheric forcing, the RUC LSM classifies precipitation types depending on the surface temperature. Then the RUC LSM updates snow water equivalent by considering snow interception by canopy similar with the Noah-MP. In calculating snow density, the RUC LSM first sets the initial values of both snow density and fresh snow density, and then corrects snow density according to the current temperature. It uses the average temperature of snow pack for snow
density and uses the surface temperature for fresh snow density. The upper limit is 400 kg/m$^3$ and the lower limit is 100 kg/m$^3$ for the snow/fresh snow density. When the surface temperature is below 258.15 K, the RUC LSM fixes fresh snow density to 100 kg/m$^3$. After all, the RUC LSM defines the average snow density of the snow pack by considering the amount of fresh snow, and updates snow depth with snow density and snow water equivalent. The fractional snow cover is calculated from snow depth and snow density.

In calculating albedo, the RUC LSM initializes the maximum snow albedo and snow-free albedo, followed by calculation of surface albedo as a function of snow depth, using roughness length, fractional snow cover, snow-free albedo and snow albedo. Lastly, by using the albedo thus obtained and the surface temperature, the RUC LSM calculates the total surface albedo assuming that the surface albedo does not change when snow temperature is below 263.15 K while it decreases when snow temperature is over 263.15 K.

### 2.1.5 The Community Land Model version 4

The Community Land Model, version 4 (CLM4; Oleson et al., 2010) is the land model component of the Community Earth System Model (Hurrell et al., 2013), version 1.0.4 (CESM1.0.4). In CLM4, many recent scientific advances in the land surface processes, especially more sophisticated representations of soil hydrology and snow processes, were incorporated to increase





the model capabilities (Oleson et al., 2010). The CLM4 has a total of 15 layers — 10 soil layers and 5 snow layers. The snow model in CLM4 is based on the parameterizations developed by Anderson (1976), Jordan (1991), and Dai and Zeng (1997). The CLM4 also starts from the new snowfall events. The CLM4 uses a critical temperature (2.5 K) to determine rain or snow. It also considers the canopy interception of precipitation by calculating the total amount of precipitation interception by vegetation,

the throughfall, and the canopy drip (Oleson et al., 2010). When the total rate of solid precipitation at the ground is calculated, a new snow depth is calculated. The CLM4 also considers the snow compaction by considering three factors: (1) temperature-dependent compaction as a result of destructive metamorphism; (2) snow load pressure-dependent compaction as a result of overburden; (3) snow ice fraction-dependent compaction resulting from snow melting (Oleson et al., 2010). Fractional snow cover is then calculated from snow density and snow depth following Toure et al. (2016) by considering the melting factor

that is set to 1. Lastly, the ground albedo is calculated from the soil and snow albedo by using fractional snow cover (Toure et al., 2016). Snow albedo is calculated depending on solar zenith angle and the model also considers albedo of the substrate underlying snow, mass concentrations of atmospheric-deposited aerosols (black carbon, mineral dust, and organic carbon), and ice effective grain size (Oleson et al., 2010).

## 2.2 Data description

In this study, the NCEP Final Operational Global Analysis (NCEP-FNL) data are used as initial and boundary conditions. To assess the results of all the LSMs, we used the Canadian Meteorological Centre (CMC) Daily Snow Depth Analysis Data, the MODIS/Tera Snow Cover Monthly L3 Global 0.05Deg CMG, Version 6, and the MODIS Bidirectional Reflectance Distribution Function (BRDF)/Albedo Product.

### 2.2.1 The NCEP-FNL data

The NCEP-FNL data are produced by the Global Data Assimilation System (GDAS) which continuously collects observations from the Global Telecommunications System (GTS) along with other sources. The NCEP-FNL shares the same data assimilation and forecast system with the Global Forecast System (GFS) but ingests more observational data. The data are prepared operationally in every six hours with a $1° \times 1°$ horizontal resolution, and includes parameters such as surface/sea-level pressure, geopotential height, temperature, sea surface temperature, soil variables, ice cover, relative humidity, horizontal wind

components, vertical motion, vorticity and ozone (see more details in https://rda.ucar.edu/datasets/ds083.2/).

### 2.2.2 The CMC Daily Snow Depth Analysis data

The CMC Daily Snow Depth Analysis data consist of the NH snow depth data, which are generated from real-time, in situ daily snow depth observation which consists of land surface synoptic (SYNOP) observation, meteorological aviation (METAR) reports, and special aviation (SA) reports from the World Meteorological Organization (WMO) information system (Brasnett,

1999). The first-guess field is obtained through a simple snow accumulation and melt model driven with the analyzed meteorological forcings from the Canadian forecast model. The snow depth observation data is interpolated to a standard 24 km polar





stereographic grid with the first-guess field. The data also include the snow water equivalent monthly data which is converted from the snow depth observation data by applying a mean monthly snow density values based on the Canadian snow course observations averaged over the snow-climate classes, defined by Sturm et al. (1995). The snow water equivalent data consists of only 9 months, excluding July, August and September. Currently, the National Snow and Ice Data Center (NSIDC) provides the

CMC Daily Snow Depth Analysis Data with a temporal coverage from 1 August 1998 to 31 December 2017. In this study, we used the data of daily snow depth and monthly snow water equivalent (see more details in https://nsidc.org/data/NSIDC-0447).

### 2.2.3 The MODIS/Terra Snow Cover Monthly L3 Global 0.05 Deg CMG, Version 6

The MODIS/Terra Snow Cover Monthly L3 Global 0.05 Deg CMG, Version 6 data are generated from the MOD10A1 snow cover data which are based on the snow-mapping algorithm using the Normalized Difference Snow Index (NDSI). Snow and

ice show very high reflections in the visible bands, and very low reflections in shortwave infrared portion of the spectrum. Therefore, NDSI is useful to detect the differences in magnitude of reflectance between the visible bands and shortwave infrared. The MOD10A1 observations then are interpolated into the 0.05 degrees climate-modeling grid (CMG) cells. The snow cover percentage then is represented by the ratio of the number of snow to the total number of land observations that are represented on the CMG cells. In this study, we used the monthly 0.5 degrees snow cover data provided by the NASA Earth

Observations (NEO), from December 2009 to May 2010 (see more details in https://nsidc.org/data/mod10cm).

### 2.2.4 The MODIS BRDF/Albedo Product

The MODIS BRDF/Albedo Product combines the multi-date, multi-angular, cloud-free, atmospherically corrected, surface reflectance observations from MODIS, both Terra and Aqua data, and the globally measured albedo from the BRDF model (Strahler et al., 1999). The albedo measurements from a satellite depends on the angle of incident light which varies with

latitude and the time of day. Therefore, these multiple albedos must be combined to provide better description of surface reflectances and absorptions of solar radiation. BRDF is a mathematical function for combining the multiple albedos by integral equation. The BRDF/Albedo parameters provide the followings: (1) the coefficients that represents BRDF of each pixel in the seven MODIS land bands (1-7); and (2) albedo measurements from not only the bands 1-7 but also the three broad bands (0.4-0.7, 0.7-3.0, and 0.4-3.0 micrometer) that are simultaneously provided by BRDF. In this study, we used the monthly 0.5

degrees albedo data provided by NEO, from December 2009 to May 2010 (see more detains in https://modis-land.gsfc.nasa. gov/brdf.html).

### 2.3 Experimental design and domain characteristics

We have conducted four experiments in order to understand how the prediction of Eurasian snow varies depending on the different choice of LSMs. All the model conditions have been fixed during the whole simulations except LSMs and the soil

layer options. For the Noah-MP that includes many microphysical options, we ran the WRF model with the default options in



the Noah-MP. For the CLM4, we calculated fractional snow cover with the model results, following Toure et al. (2016), as:

$$F_s = \tanh\left\{\frac{H_s}{2.5 Z_{0m,g}[\min(\rho_s, 800)/\rho_{s,new}]^m}\right\} \tag{3}$$

where $F_s$ is fractional snow cover, $H_s$ is snow depth, $\rho_s$ is snow density, $Z_{0m,g} = 0.01$ is the momentum roughness length for soil, $\rho_{s,new} = 100 \text{ kg/m}^3$ is assumed fresh snow density, and $m = 1$ is melting factor. Among the LSMs coupled to WRF,
we excluded the Thermal Diffusion (TD) scheme (Dudhia, 1996) and the Pleim-Xiu LSM (Pleim and Xiu, 1995), since they do not consider the snow depth in their snow scheme. The spin-up time is set to 6 months, from 1 June 2009 to 30 November 2009.

Figure 1 shows the characteristics of the study domain — the terrain height (Fig. 1a) and the dominant vegetation category (Fig. 1b). The highest altitudes over the whole domain, including the Himalayas and the Tibetan Plateau, appear in the area of
75 °E − 95 °E and 23 °N − 35 °N. Many mountain ranges, such as the Pamir Mountains, the Hindu Kush, the Karakoram, the Tien Shan and the Altai Mountains, extend from the Himalayas and the Tibetan Plateau. Therefore, the terrain height of the northern and eastern parts of the study domain appears to be relatively lower than that of the central and southwestern parts, except the Ural Mountains which are lined along 60°E between 55 °N and 65°N. Deciduous and evergreen forests appear along the area between 55 °N and 65 °N, which are represented by vegetation category from 11 to 14. In the area below 50 °N, the
grassland type (category 7) is prominent. Generally, the desert area has land types of barren/sparsely vegetated (category 19), grassland (category 7), mixed shrubland/grassland (category 9), and dryland cropland and pasture (category 2).

### 2.4 Validation methods

We have assessed the results from each LSM, using several statistical parameters — the mean difference, the relative bias, the root mean square difference, and the Pearson's correlation coefficient. For the statistical assessment, we interpolated the model
grids to the observation grids by using the inverse distance weighting (IDW) method as:

$$\hat{Z}_o = \frac{\sum_{i=1}^{n} w_i z_i}{\sum_{i=1}^{n} w_i} \tag{4}$$

$$w_i = \frac{1}{d_{o,i}} \tag{5}$$

where $n$ is the number of points, $\hat{Z}_o$ is the interpolated value, $z$ is the variable value at the grid point $i$, $w$ is the weight, and $d_{o,i}$ is the distance between the grid points of $o$ and $i$.
The mean difference (MD) is calculated by

$$(MD)_{i,j} = M_{i,j} - O_{i,j} \tag{6}$$

where $M$ is the model value, $O$ is the observation, $t$ is time, $i$ and $j$ are grid points. We use the mean difference to see how each LSM shows quantitative differences with the observation.

The relative bias (RB) is calculated by

$$RB = \frac{M_{i,j}}{O_{i,j}} - 1. \tag{7}$$





We excluded the grid point where the observation value is zero. Through the relative bias we can see how each result is biased.

The root mean square difference (RMSD) is calculated as:

$$RMSD = \sqrt{\overline{(M_{i,j} - O_{i,j})^2}} \qquad (8)$$

where the overbar represents the mean value. The RMSD prevents the canceling-out effect within the reverse signs, so we use
RMSD to see the absolute quantitative differences in the results of LSMs with the observation.

The Pearson's correlation coefficient ($r$) is calculated as:

$$r = \frac{\sum_{i=1}^{n} \sum_{j=1}^{n} (M_{i,j} - \overline{M})(O_{i,j} - \overline{O})}{\sqrt{\sum_{i=1}^{n} \sum_{j=1}^{n} (M_{i,j} - \overline{M})^2} \sqrt{\sum_{i=1}^{n} \sum_{j=1}^{n} (O_{i,j} - \overline{O})^2}} \qquad (9)$$

where $i$ represents the every single grid points, $\overline{M}$ is the average of model values and $\overline{O}$ is the average of observations. By using the correlation coefficient, we evaluate each LSM to see how the results spatially and quantitatively match well with the
observations.

## 3    Results

In our results, the two-dimensional snow-related fields are averaged for a period from December 2009 to May 2010.

### 3.1    Snow depth

Figure 2 shows snow depth produced by WRF using the four LSM options (Noah LSM, RUC LSM, Noah-MP, and CLM4). All
the LSMs show reasonable results in snow depth: in general, snow depths tend to be deeper over the regions of higher altitude and latitude. Especially snow depths show the highest values near the Pamir Mountains, the Hindu Kush, the Karakoram, the Tien Shan and the Altai Mountains, which are among the highest elevations over the domain. It is also noticed that all the LSMs bring about relatively high snow depths at the downstream sides of the water bodies, especially when combined with some high terrains — e.g., the Black Sea and the Caucasus Mountains, Lake Baikal and the Yablonovy Range, and the East Sea/Sea of
Japan and the high-mountain areas in Japan (e.g., Hokkaido and Japanese Alps). During the winter time, snow is usually formed by the western or northwestern flows in these areas. With regard to the inland water bodies, when the wind blows over the relatively warmer lake surface, moisture is carried along the wind to form clouds under adequate conditions and sometimes to make the lake-effect snowfall (e.g., Braham and Dungey, 1995; Liu and Moore, 2004; Obolkin and Potemkin, 2006). Then snow generally piles up at the downstream side of the lake. The LSMs in this study simulate the lake-effect snowfall reasonably
well by estimating relatively higher snow depth over the downstream side (e.g., southeast area) of Lake Baikal. However, each model estimates the snow depth differently due to different physical processes and parameterizations. The Noah LSM tends to predict the highest snow depth (Fig. 2a) while the Noah-MP trends to predict the lowest snow depth (Fig. 2c) during both the winter and spring seasons. Table 2 depicts that all LSMs result in higher snow depth with smaller standard deviation during winter compared to spring. The standard deviation throughout the seasons indicates that the variability of snow depth during
spring is larger than that during winter as the snow melting effect becomes larger.





Figure 3 shows the time series of the domain weighted average snow depths in comparison between the LSM results and the CMC observations. The Noah LSM largely overestimates snow depth in most of the winter/spring period, even from the beginning of December, whereas the other LSMs show similar values with observation. In Fig. 4, the Noah LSM shows lower total precipitation amount (Fig. 4a) than the Noah-MP and CLM4 throughout the whole period. However, the Noah LSM

predicts the highest amount of snowfall (Figs. 4b and 4c). All the LSMs, except the Noah LSM, consider the canopy-intercepted amount in calculating new snowfall. Given that the snowfall difference increases in spring as the growth of vegetation increases, the large difference in the Noah LSM possibly comes from the lack of the canopy-interception effect in calculating new snowfall. From mid-December, the RUC LSM and CLM4 also start to show prominent differences from observation while the Noah-MP shows the best matching result with observation during the study period. Considering that the snowfall amount from

each LSM during winter are mostly similar (Fig. 4b and 4c) and the surface temperature is remained below the melting point (Fig. 5), the specific snow compaction parameterization of each LSM resulted in differences in snow depths. In spring, the observed snow depth starts to decrease from late March. The peak periods of snow depth seem to match well for all the LSMs, except the Noah LSM which shows the peak period near the end of February. However, all the LSMs tend to overestimate the snow melting effect during April and May because of the rapid increase of surface temperature and the lower amount

of snowfall compared to the previous months. Since the Noah-MP predicts relatively high surface and air temperature while predicting the lowest snowfall, it shows the lowest value of snow depth during the melting season as in the observation.

Over most of the study area, all the LSMs generally tend to overestimate snow depth (Fig. 6). The Noah-MP shows the best fit to the observation with the highest correlation coefficient (0.64) and the Noah LSM shows the largest overestimation with the lowest correlation coefficient (0.48). In addition, the range of difference between the observation and each LSM

seems to become larger as snow depth deepens. Figure 7 shows the relative bias of snow depth over the study area. All the LSMs commonly represent negative biases near the mountain area, and relatively low biases appear along the evergreen broadleaf/needleleaf and the dry cropland/pasture areas (see Fig. 1b). The Noah LSM and the RUC LSM show positive biases mostly over the high latitudes of Eurasia while the CLM4 and the Noah-MP show negative biases partly on the region where the deciduous broadleaf/needleleaf and the evergreen broadleaf/needleleaf forests exist. Additionally, the CLM4 and the Noah-MP

show much lower bias than the other LSMs, especially over the high latitude deciduous broadleaf forest region. In particular, very high biases appear along the grassland (40°N-50°N) in all the LSMs. The grassland areas include the western/eastern steppe, the Gobi Desert, the Mongolian Plateau, and the Horqin Desert.

In the snow depth estimation, the Noah-MP shows the best results in time- and space-averaged values of mean difference (-0.01), relative bias (0.70), and RMSD (0.14) (see Table 3).

## 3.2 Fractional snow cover

Fractional snow cover is highly related with snow depth since the former is usually calculated by using the latter. It is very important to predict the snow coverage on the land surface, since it is directly linked to the surface albedo which is of the utmost importance in considering the surface energy budget. The overall spatial pattern of fractional snow cover over the study domain (Fig. 8) seems similar to that of snow depth (cf. Fig. 2). In all the LSMs, fractional snow cover shows relatively high



values at the high latitude and mountain areas. In particular, the spatial patterns show similarity between the Noah LSM and the RUC LSM and between the CLM4 and the Noah-MP, respectively.

These similarities are also represented in the time series of daily variations and the overall pattern of fractional snow cover (Fig. 9a). Note that the Noah LSM directly derives fractional snow cover from snow water equivalent and its threshold while

the other LSMs derive fractional snow cover by using snow depth and snow density. Especially, the CLM4 and the Noah-MP additionally consider ground roughness length and the melting factor for estimation of fractional snow cover by using a function of hyperbolic tangent. The more sophisticated schemes in the CLM4 and the Noah-MP make the distinctive differences compared to the Noah LSM and the RUC LSM. Furthermore, in the CLM4 and the Noah-MP, snow density is derived from the atmospheric temperature and the freezing/melting point temperature; thus both models represent variations similar to the at-

mospheric temperature patterns (cf. Fig. 5a). Although the patterns are similar, the Noah-MP always estimates lower fractional snow cover than the CLM4 does. The differences between the Noah-MP and the CLM4 may be due to the higher estimation of snow depth in the CLM4 with the lower melting factor. In Fig. 9b, the peak of observation appears at the end of December 2009. The peak periods of the CLM4 and the Noah-MP match well with the observation while those of the Noah LSM and the RUC LSM are delayed, at the end of January 2010. The Noah LSM shows the highest fractional snow cover during the study

period except December 2009. Since the Noah LSM directly calculates fractional snow cover from snow water equivalent, the large amount of snowfall during this period results in high fractional snow cover by showing similar patterns between fractional snow cover and snowfall rate. On the contrary, the RUC LSM shows relatively low fractional snow cover during the winter time (see Fig. 8b). Note that we exclude the area of null fractional snow cover (see the brown-shaded area in Fig. 8) in calculating the domain-averaged fractional snow cover; thus this low fractional snow cover can be attributed to the snow cover

area predicted by the RUC LSM. In the snow melting season, all the LSMs show much larger differences with the observation except the Noah-MP.

Figure 10 shows the relative bias of time-averaged fractional snow cover. The domain of the relative bias has been reduced because of the different map projection and spatial resolution between the model and the observation. The most prominent feature is that all the LSMs have strong positive bias over the area from the Tibetan Plateau to the Mongolian Plateau where

the grassland type is dominant, except the RUC LSM. The RUC LSM shows relatively high negative bias over that region with partly representing strong positive bias at some areas such as the boundary region between the Takla Makan Desert and the Kunlun Mountains, Mongolian Plateau and the Horqin Desert. Unlikely to the other LSMs, the Noah-MP shows the negative bias over most of the domain at 50°N and above. Figure 11 depicts the mean difference between the model fractional snow cover and the observation, and more specifically shows the difference of features depending on each LSM over the regions of

large biases. All the LSMs tend to overestimate fractional snow cover over the boundary between the Takla Makan Desert and the Kunlun Mountains. The high mean differences also occur over the Mongolian Plateau and the Horqin Desert, similarly to snow depth. Over the grassland area along 45°N including Altai Mountains (i.e., the northeastern part of the Mongolia), all the LSMs commonly underestimate and the RUC LSM represents the largest difference with the observation. The Noah-MP underestimates fractional snow cover all over the high latitude region where the other LSMs overestimate it.





Overall, we evaluate that the Noah-MP estimates fractional snow cover most accurately with the lowest MD (0.02) and RMSD (0.21) (see Table 3. In addition, the Noah-MP shows the best corresponding result with the observation in time series during the study period. Although the RUC LSM shows the lowest value in relative bias, this effect seems to be cancelled out by the strong negative bias. In addition, the RUC LSM shows higher values in MD and RMSD than the Noah-MP does.

Therefore, we regard the Noah-MP as the best performing LSM in predicting the fractional snow cover.

### 3.3  Surface albedo

The surface albedo is also highly related with snow since LSMs calculate the surface albedo based on the snow-free albedo and the snow albedo. The surface albedo directly influences the surface energy budget, and snow has a huge impact on the surface albedo by reflecting more solar radiation than other types of the land surface. In calculating the surface albedo, all the LSMs

use fractional snow cover to give the weight to the snow albedo and the snow-free albedo. Therefore, the spatial pattern of surface albedo (Fig. 12 is similar to that of both snow depth and fractional snow cover. However, unlike the previous variables, the surface albedo shows distinctive boundaries between different kinds of vegetation, especially in the CLM4 and the Noah-MP. As in fractional snow cover, the Noah-MP generally estimates relatively low values over the study domain compared to the other LSMs. In contrast, the Noah LSM estimates high surface albedo throughout the study area, especially over the

barren/sparsely vegetated area even with no snow. Of all the LSMs, the CLM4 demonstrates the most sensitive estimation of surface albedo on the vegetation type. For example, along the latitudes between 55°N and 65°N where the vegetation is categorized as the evergreen forest and the deciduous forest, the CLM4 estimates albedo distinctively between the broadleaf forest and the needleleaf forest.

In the time series of domain-averaged surface albedo (Fig. 13a), each LSM shows similar fluctuation of the surface albedo in

phase with fractional snow cover; when fractional snow cover increases/decreases, the surface albedo also increases/decreases with smaller amplitude. Therefore, the peak periods of surface albedo match well with those of fractional snow cover in all the LSMs (cf. Figs. 9b and 13b). As shown in the previous results, the Noah LSM shows the highest value of surface albedo throughout the study period, representing the largest differences with the observation. However, the surface albedo from the RUC LSM shows different aspect, compared to fractional snow cover, by representing more variations. The RUC LSM uses the

surface temperature in calculating the surface albedo; thus resulting in more fluctuations in the surface albedo, similar to the surface temperature pattern. All the LSMs tend to miss the declining trend during February and March; however, the Noah-MP catches up the observation during the remaining spring season.

Figure 14 shows the relative bias of the time-averaged surface albedo over the study domain. The spatial pattern is similar to the fractional snow cover counterpart though the bias amplitude has been reduced. In the areas where the LSMs overestimate

fractional snow cover, the surface albedos also tend to be overestimated since the latter is proportional to the former. Commonly, all the LSMs highly overestimate the surface albedo over the high mountain regions stretched from the Pamir Mountains. In addition, all the LSMs significantly represent high positive relative bias over the evergreen broadleaf and needleleaf forest areas along 60°N, especially in the CLM4. The CLM4 also depicts negative mean differences over the shrubland and the grassland,





represented as category 7, 8 and 9 in Fig. 1b. It is also noteworthy that all the LSMs show relatively high positive bias over the evergreen broadleaf forest areas along the coastal line.

Figure 15 represents the mean difference of surface albedo. The MD pattern of surface albedo is similar to that of fractional snow cover (cf. Fig. 11): positive (negative) MD of fractional snow cover corresponds to positive (negative) MD of surface albedo. As the albedo is directly affected by the fractional snow cover, it is confirmed that the higher accuracy of fractional snow cover results in higher accuracy of surface albedo. The Noah LSM (Fig. 15a) predicts distinctively high surface albedo compared to the others. This has direct impact on predicting lower surface temperature in winter and spring because the surface with higher albedo reflects more radiation.

Overall, the Noah-MP shows the best result in estimating the surface albedo with the lowest value of MD (0.06), relative bias (0.28) and RMSD (0.13) (see Table 3). In time series of the domain-averaged surface albedo, the Noah-MP also shows the most corresponding result with the observation.

## 4    Discussions and conclusions

In this study, we have investigated the performance of several land surface models (LSMs), coupled with the Weather Research and Forecasting (WRF) model version 3.6.1, in terms of predicting the Eurasian snow. The LSMs include the Unified Noah LSM, the Noah LSM with multiparameterization options (Noah-MP), the Rapid Update Cycle (RUC) LSM, and the Community Land Model, version 4 (CLM4). The study aims at achieving the following goals: 1) to understand the structure of each LSM, focusing on the snow schemes; 2) to analyze why the snow aspects show different results for different LSMs; 3) to evaluate each LSM and suggest which LSM performs the best in predicting the Eurasian snow; and 4) to suggest how to further improve the model performance.

In terms of the first goal, the general structure of snow schemes can be summarized as the followings: 1) all LSMs, except the Noah LSM, calculate snow water equivalent, snow density, snow depth, fractional snow cover, and albedo sequentially from precipitation; 2) the Noah LSM calculates snow density, snow depth, and fractional snow cover from snow water equivalent that is derived by precipitation, rather than sequential calculation; and 3) the Noah-MP and the CLM4 show relatively complex schemes compared to the Noah LSM and the RUC LSM by considering more environmental variables in calculating each snow-related variable.

For the second goal, the snow aspects from the LSMs are analyzed and discussed in terms of: 1) snow depth, 2) fractional snow cover, and 3) albedo. Estimation of snow depth is highly influenced by the atmospheric forcings such as the amount of total precipitation, atmospheric temperature and surface temperature. By using these forcings, the LSMs calculate the amount of new snowfall. Then they compile snow over the land surface with their own snow schemes which consider the snow interception processes by the canopy, snow compaction, and snow melting. The Noah LSM shows much larger difference with the observation than the other LSMs while the Noah-MP shows the best matching result. The Noah LSM lacks snow compaction due to just one snow layer with high amount of snowfall, which leads to deficiency in estimating snow depth. On the other





hand, the Noah-MP and the CLM4 show relatively good results compared to the other LSMs by delicately considering snow compaction, which includes the effects of destructive metamorphism, overburden, and snow melting.

Based on these results, the important factors for predicting snow depth can be summarized as the followings: 1) the amount of snowfall at the ground surface, 2) atmospheric and surface temperature, 3) snow interception by the canopy, and 4) snow compaction. Fractional snow cover is usually calculated from snow depth and snow density, except the Noah LSM in which it is calculated directly from snow water equivalent. Therefore, the accuracy of fractional snow cover estimation is strongly dependent on the accuracy of snow depth estimation. As the Noah LSM estimates high amount of snowfall, it also estimates high fractional snow cover over the domain throughout the study period, resulting in the lowest accuracy in predicting fractional snow cover. The Noah-MP and the CLM4 show high variations similar to the trend of atmospheric temperature during the study period since they use atmospheric temperature in calculating snow density. The RUC LSM also uses atmospheric temperature in calculating snow density; however, limiting the maximum snow density (i.e., to 400 kg/m$^3$) can influence the RUC LSM to have less variations and lower fractional snow cover during the winter snow accumulation period.

The surface albedo is highly affected by snow, especially in winter when a large portion of the land surface is covered by the snow. With the high albedo, snow influences the surface energy budget, which is of the utmost importance in the climate and weather systems. The surface albedo is calculated by using fractional snow cover to consider the effect of snow albedo; thus, the accurate estimation of fractional snow cover is considered to be one of the most important factors in predicting the surface albedo. In this study, we focused on the variations of surface albedo depending on the variations of fractional snow cover. All the LSMs show similarity in patterns between surface albedo and fractional snow cover. The RUC LSM shows relatively high variations in the time series of surface albedo than fractional snow cover since it uses surface temperature in calculating the surface albedo. Especially the snow albedo is highly influenced by the new snowfall rather than the old snow. Therefore, the trend of the time series of surface albedo seems to be impacted by the amount of the new snowfall. In addition, all the variables show similar spatial patterns. Usually, over the area where snow depth is overestimated, fractional snow cover is also overestimated, which consequently resulted in higher surface albedo than the observation. All the LSMs show high bias over the mountainous regions with high altitude, which is considered to be caused by the followings: 1) the lack of accurate atmospheric forcings such as the amount of precipitation and atmospheric temperature; 2) the lack of accurate estimation of snow compaction in the snow schemes; and 3) the lack of high spatial resolution to represent accurate values over the mountainous regions where the amount of snow can rapidly increase depending on the altitude. We also note that all the LSMs tend to show low accuracy especially over the open grassland, among all the vegetation types. Usually, over the open grassland, redistribution of snow by wind can make high variations in the snow related aspects. Therefore, snow depth can highly vary with sensitive dependence on the daily variation of weather, consequently influences fractional snow cover and the surface albedo. As a result, the high bias can also result from differences in time- and space-resolution of LSMs and the observations.

To achieve the third goal, we have evaluated all the LSM results by using the mean difference (MD), root mean square difference (RMSD), relative bias (RB), and Pearsons' correlation. The performance of LSMs in terms of the accuracy in predicting all three parameters, i.e., the Eurasian snow depth, fractional snow cover and surface albedo, can be listed in order



as 1) Noah-MP, 2) CLM4, 3) RUC LSM, and 4) Noah LSM. In predicting fractional snow cover, the RUC LSM shows the lowest RB among all the LSMs and lower MD than the CLM4 does; however, it predicts low fraction of snow cover over the low latitude area, which caused negative MD and RB over a large area. Since the RUC LSM shows higher RMSD and lower correlation compared to the Noah-MP and the CLM4, the large area with the negative MD and RB is considered to show lower

values than the other LSMs through the cancelling-out effect.

Furthermore, we also evaluated the spatial pattern of each LSM, using a Taylor diagram (see Fig. 16), which represents the spatial correlations between the model-produced snow aspects (i.e., snow depth, fractional snow cover and albedo) and the corresponding observations. It is noteworthy that, in estimating quantitative snow depth (i.e., category 1), the Noah LSM (red dot) shows higher spatial correlation than the RUC LSM and the CLM4. This is attributed to the overestimation of snow

depth by the Noah LSM mainly due to the lack of snow compaction and the overestimation of snowfall amount. Snow depth shows the highest spatial correlation coefficients followed by fractional snow cover and albedo. Fractional snow cover shows the largest range of differences in spatial correlation ($0.39-0.55$) for the choice of LSMs. The surface albedo is one of the ultimate reasons why snow is important during the winter and spring time; however, it shows the lowest correlation among the three snow aspects in all the LSMs.

As the last goal of this study, we suggest that the Eurasian snow estimation in the LSMs coupled to the WRF model can be further improved by the followings: 1) using more advanced and sophisticated microphysics schemes (e.g., WSM6 that considers the graupel) can improve the amount of winter precipitation; 2) to enhancing the snow compaction processes with more accurate snow density will improve the LSM performance over the high latitude/altitude where snow stays throughout the year; 3) treating the vegetation effect more delicately can enhance the model performance especially over the open grassland;

and 4) applying the optimal parameter estimation technique, e.g., using the genetic algorithm (Lee et al., 2006; Yu et al., 2013; Hong et al., 2014, 2015), can bring about further improvement of prediction.

Overall, all the LSMs show reasonable results. In particular, the Noah-MP predicts the Eurasian snow most accurately compared to the observations in terms of both the temporal/spatial patterns and the quantitative aspects. Estimation of the Eurasian snow can be further improved by applying proper microphysics scheme, and improving environmental parameters by

considering such as vegetation, snow compaction, and snow melting effects.

*Acknowledgements.* This research was supported by Basic Science Research Program through the National Research Foundation of Korea (NRF) funded by the Ministry of Education (2018R1A6A1A08025520).

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



**Table 1.** WRF Version 3.6.1 Configuration

| WRF Version 3.6.1 Configuration | | References |
|---|---|---|
| Simulation period | 1 June 2009 - 31 August 2010 | |
| Dynamic core | ARW | Skamarock et al. (2008) |
| Horizontal resolution | 30 km | |
| Vertical levels | 30 (model top at 50 hPa) | |
| PBL/turbulence parameter | YSU | Hong et al. (2006); Hong and Kim (2008) |
| Microphysical parameter | WSM3 | Hong and Lim (2006) |
| Radiation (Shortwave) | Dudhia | Dudhia (1989) |
| Radiation (Longwave) | RRTM | Mlawer et al. (1997) |
| Convective parameter | Kain-Fritsch (new Eta) scheme | Kain and Fritsch (1993); Kain (2004) |
| Land Surface Models (LSMs) | Unified Noah LSM | Chen et al. (1996); Koren et al. (1999) |
| | Noah-MP | Niu and Yang (2007); Niu et al. (2011) |
| | RUC LSM | Benjamin et al. (2004) |
| | Community Land Model version 4 (CLM4) | Anderson (1976); Jordan (1991); Dai and Zeng (1997) |

**Table 2.** Snow depth (in m), fractional snow cover and albedo from the four LSMs.

| | | Snow depth [m] | | Fractional Snow cover [-] | | Surface albedo [-] | |
|---|---|---|---|---|---|---|---|
| | | Avg | Std | Avg | Std | Avg | Std |
| Noah LSM | Total | 0.23 | 0.106 | 0.36 | 0.131 | 0.38 | 0.068 |
| | DJF | 0.26 | 0.067 | 0.41 | 0.059 | 0.41 | 0.032 |
| | MAM | 0.30 | 0.108 | 0.30 | 0.135 | 0.35 | 0.069 |
| RUC LSM | Total | 0.19 | 0.080 | 0.30 | 0.108 | 0.31 | 0.079 |
| | DJF | 0.19 | 0.057 | 0.34 | 0.061 | 0.35 | 0.059 |
| | MAM | 0.19 | 0.077 | 0.27 | 0.110 | 0.27 | 0.065 |
| Noah-MP | Total | 0.16 | 0.077 | 0.34 | 0.165 | 0.30 | 0.097 |
| | DJF | 0.17 | 0.051 | 0.41 | 0.079 | 0.36 | 0.054 |
| | MAM | 0.16 | 0.079 | 0.26 | 0.156 | 0.25 | 0.075 |
| CLM4 | Total | 0.14 | 0.082 | 0.29 | 0.193 | 0.25 | 0.082 |
| | DJF | 0.16 | 0.048 | 0.40 | 0.096 | 0.30 | 0.052 |
| | MAM | 0.12 | 0.083 | 0.18 | 0.162 | 0.21 | 0.066 |




**Table 3.** The mean difference (MD), relative bias (RB), root mean square difference (RMSD), and spatial correlation coefficient (SCC) of snow depth, fractional snow cover, and albedo for the four LSMs.

| | | Noah LSM | RUC LSM | Noah-MP | CLM4 |
|---|---|---|---|---|---|
| Snow depth | MD | 0.08 | 0.04 | 0.02 | -0.01 |
| | RB | 2.10 | 1.08 | 0.99 | 0.70 |
| | RMSD | 0.21 | 0.18 | 0.16 | 0.14 |
| | SCC | 0.66 | 0.62 | 0.59 | 0.66 |
| Fractional snow cover | MD | 0.11 | 0.04 | 0.08 | 0.02 |
| | RB | 3.63 | 0.93 | 2.19 | 2.75 |
| | RMSD | 0.25 | 0.24 | 0.22 | 0.21 |
| | SCC | 0.44 | 0.39 | 0.48 | 0.55 |
| Surface albedo | MD | 0.21 | 0.14 | 0.13 | 0.06 |
| | RB | 0.91 | 0.56 | 0.51 | 0.28 |
| | RMSD | 0.25 | 0.20 | 0.22 | 0.13 |
| | SCC | 0.28 | 0.34 | 0.40 | 0.39 |

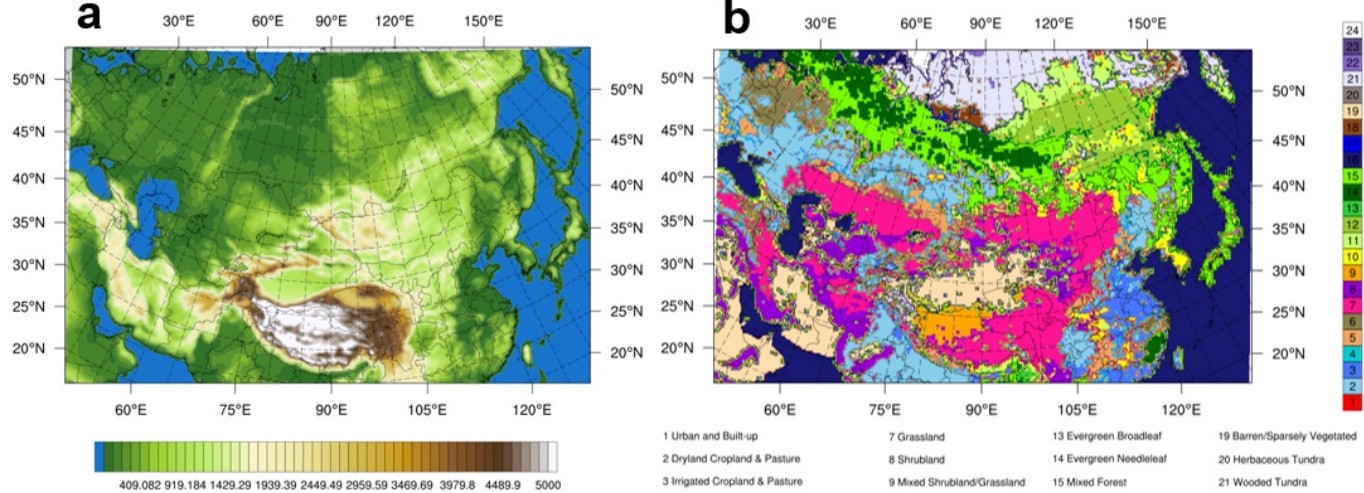

**Figure 1.** Computational domain: (a) the terrain height (in m) and (b) the dominant vegetation types categorized by the USGS 24-category Land Use Categories.





**Figure 2.** Snow depth (in m) for (a) Noah LSM, (b) RUC LSM, (c) Noah-MP, and (d) CLM4, averaged for a period from December 2009 to May 2010.





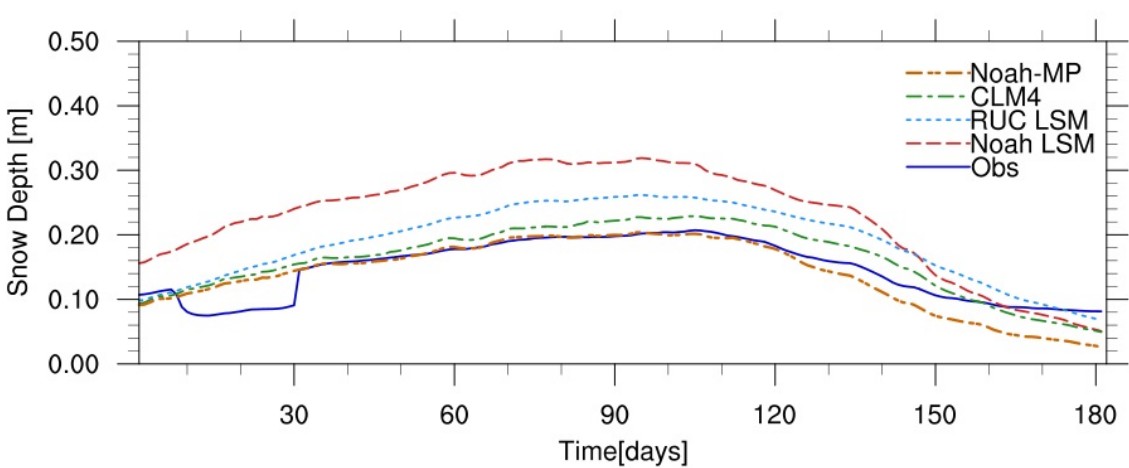

**Figure 3.** The daily time series of the area-averaged snow depth for the four LSMs and CMC snow depth observation from 1 December 2009 to 31 May 2010. The spin-up time and the periods with low snow depth were excluded.





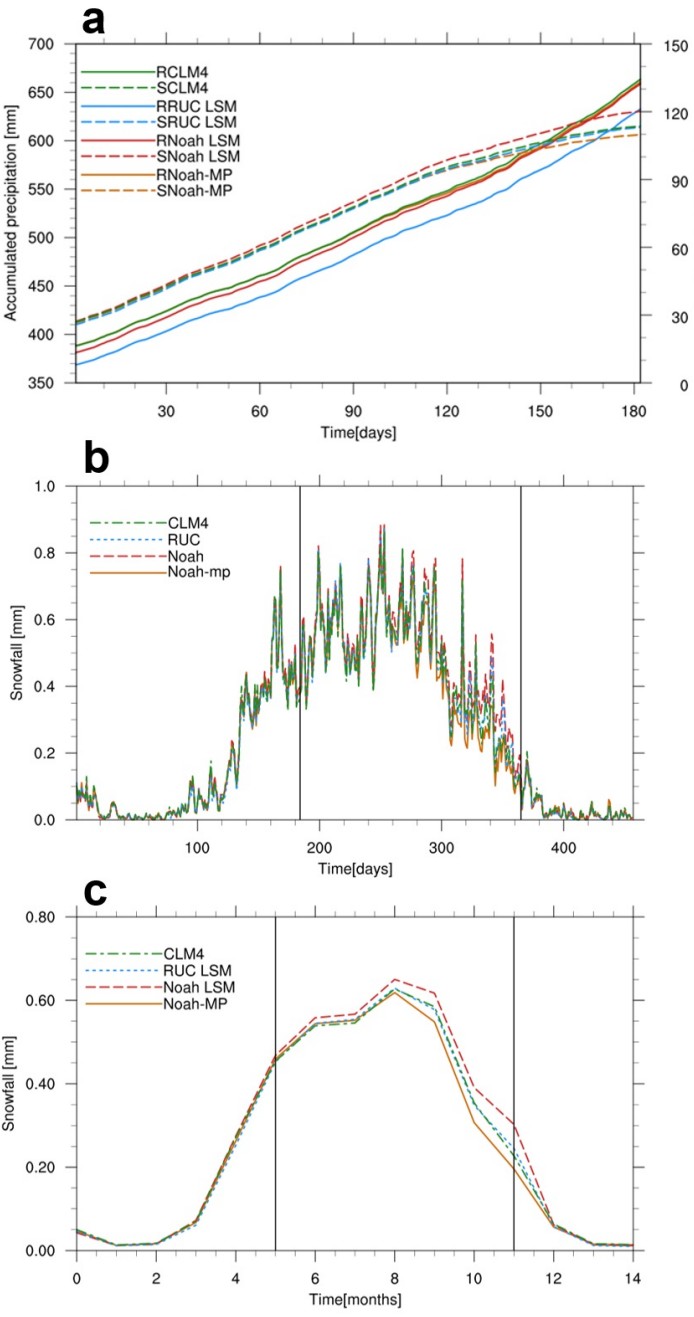

**Figure 4.** The amount of (a) accumulated total precipitation and snowfall during the period of large snowfall amount (December 2009 to May 2010), (b) daily snowfall, and (c) monthly snowfall during the simulation period (June 2009 to August 2010). In (a), 'R' and 'S' in front of the model names indicate total precipitation and snowfall, and are represented by solid and dashed lines respectively. Two vertical lines represent the start of the winter period (left) and the end of the spring period (right), respectively.



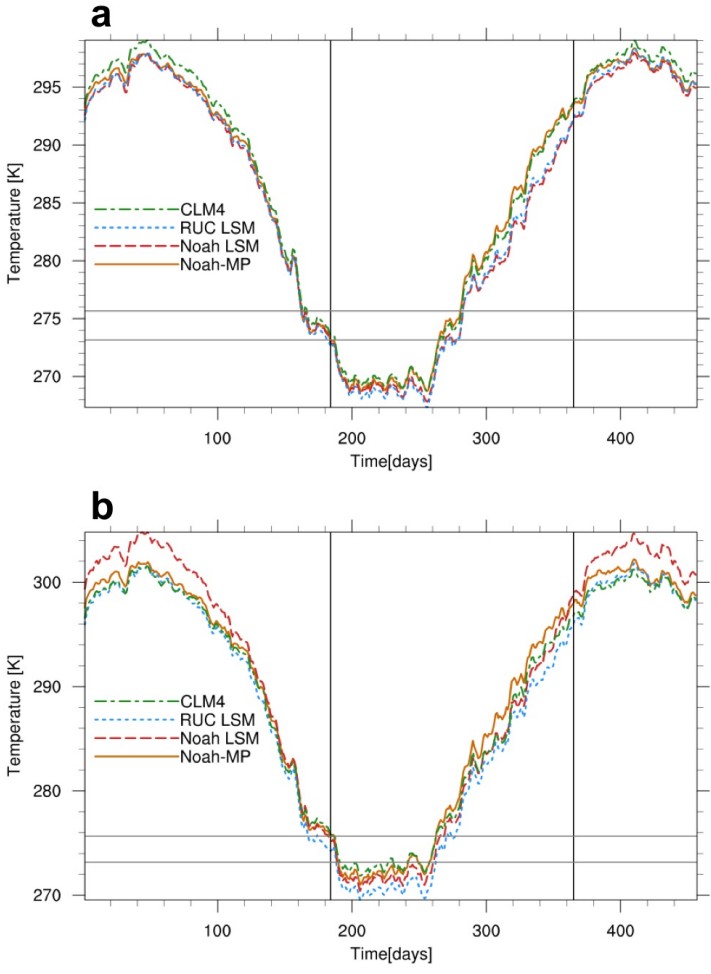

**Figure 5.** The domain averaged values of (a) 2m air temperature and (b) surface temperature from the four LSMs during the simulation period. The two vertical lines represent the start of winter (left) and the end of spring (right), respectively. The horizontal lines represents temperature values of 273.16 K (lower) and 273.16+2.5 K (upper), respectively.





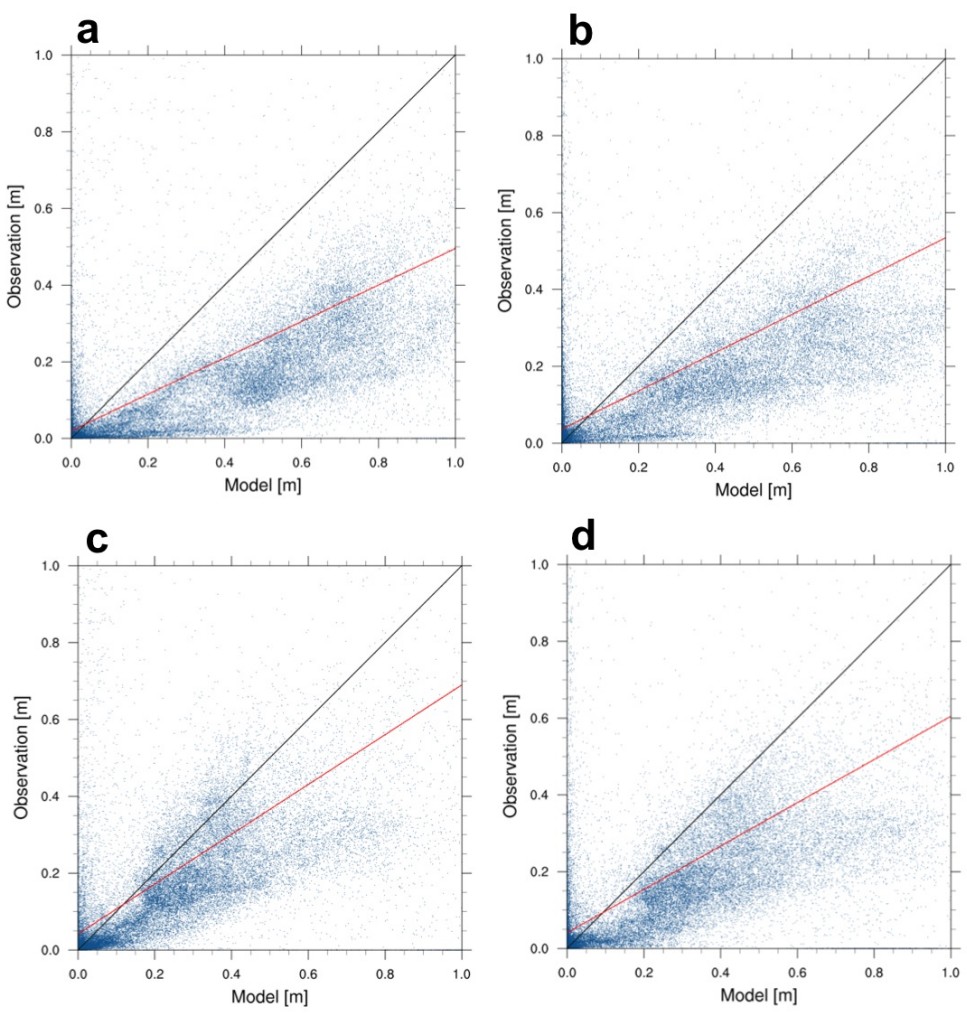

**Figure 6.** The scatter plot of time averaged snow depth for (a) Noah LSM, (b) RUC LSM, (c) Noah-MP, and (d) CLM4. Snow depth is averaged for 6 months, from December 2009 to May 2010, and is plotted (i.e., dotted) for each grid point over the domain. The horizontal axis represents the model values and the vertical axis depicts the observation values. The red solid line describes the regression line.



**Figure 7.** Same as in Fig. 2 but for the relative bias of snow depth.





**Figure 8.** Same as in Fig. 2 but for fractional snow cover. The brown-shaded area indicates null fractional snow cover.



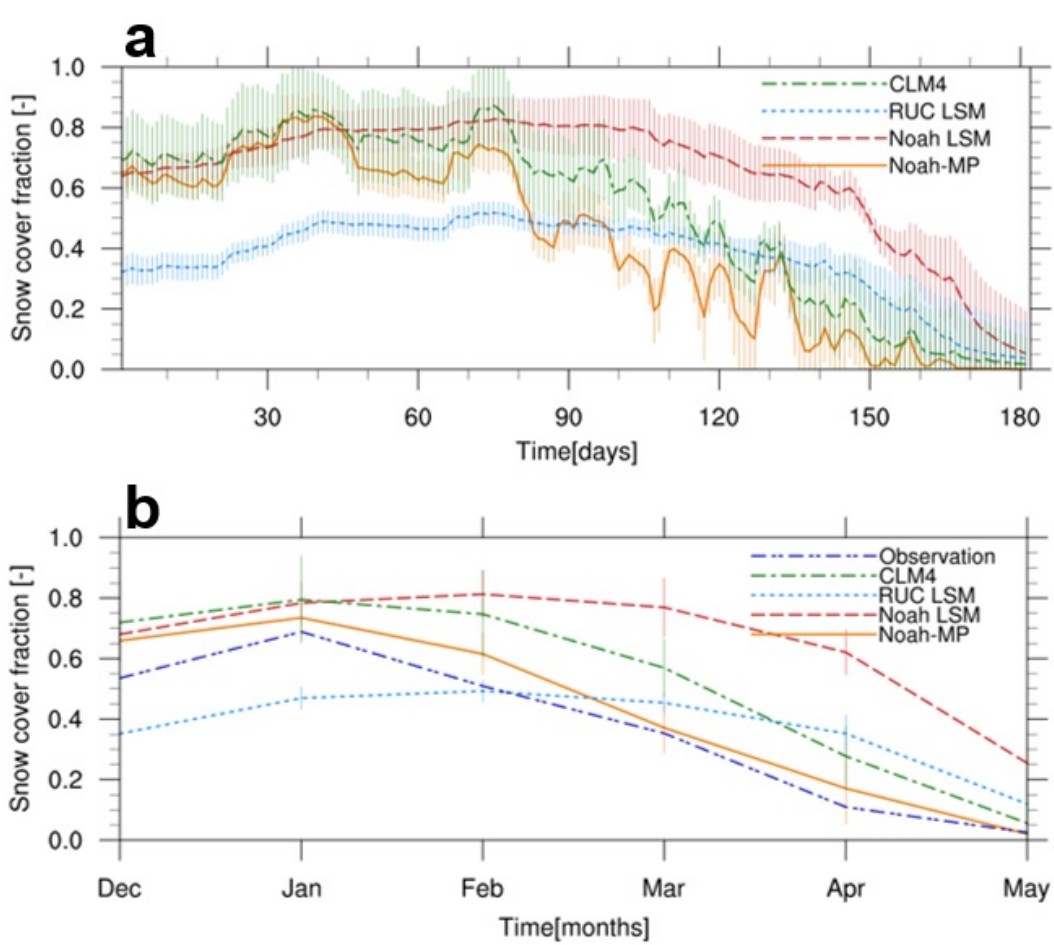

**Figure 9.** The domain-averaged time series of (a) daily fractional snow cover and (b) monthly averaged fractional snow cover. Observations are shown in (b). The time series are plotted from 1 December 2009 to 31 May 2010, excluding the spin-up time and the periods with low snow depth.

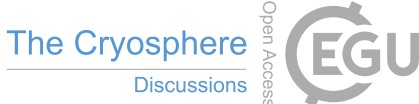



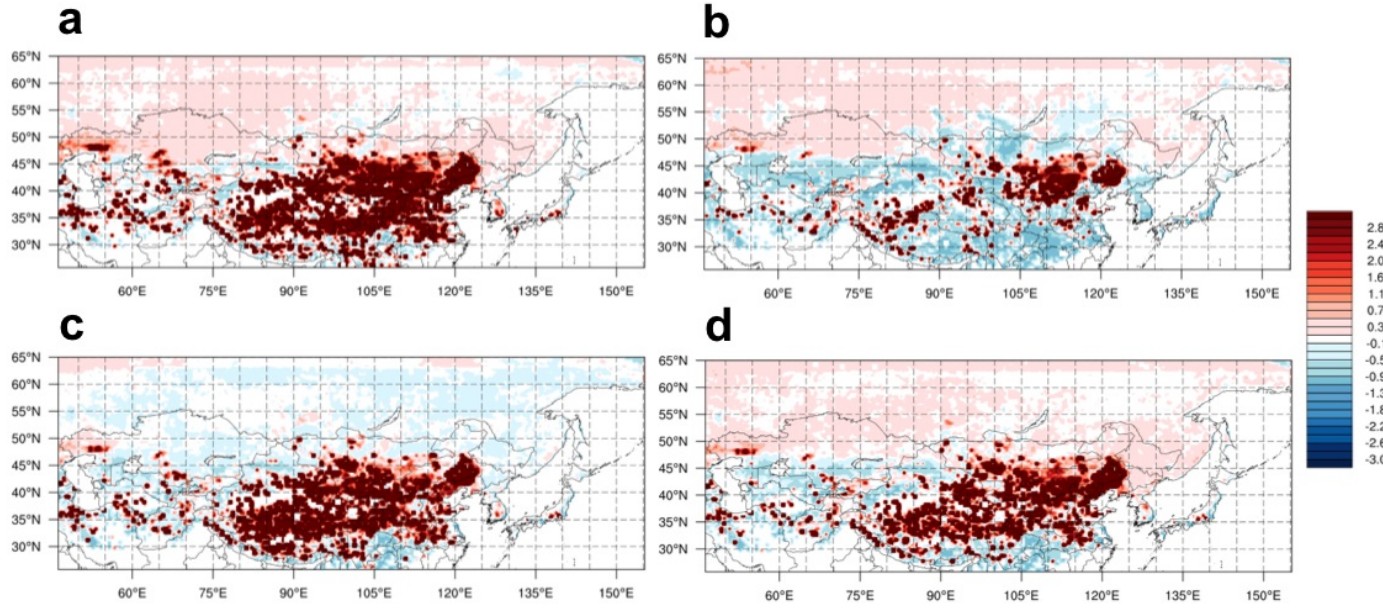

**Figure 10.** Same as in Fig. 2 but for the relative bias of fractional snow cover.

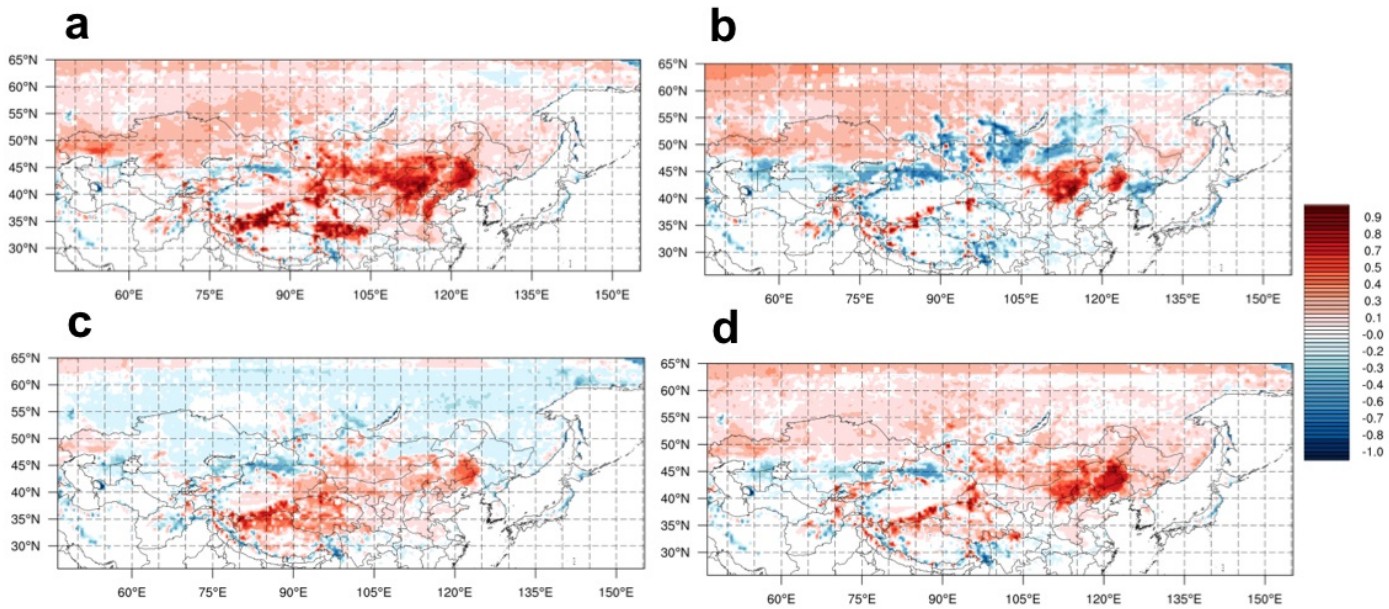

**Figure 11.** Same as in Fig. 2 but for the mean difference of fractional snow cover.





**Figure 12.** Same as in Fig. 2 but for surface albedo.



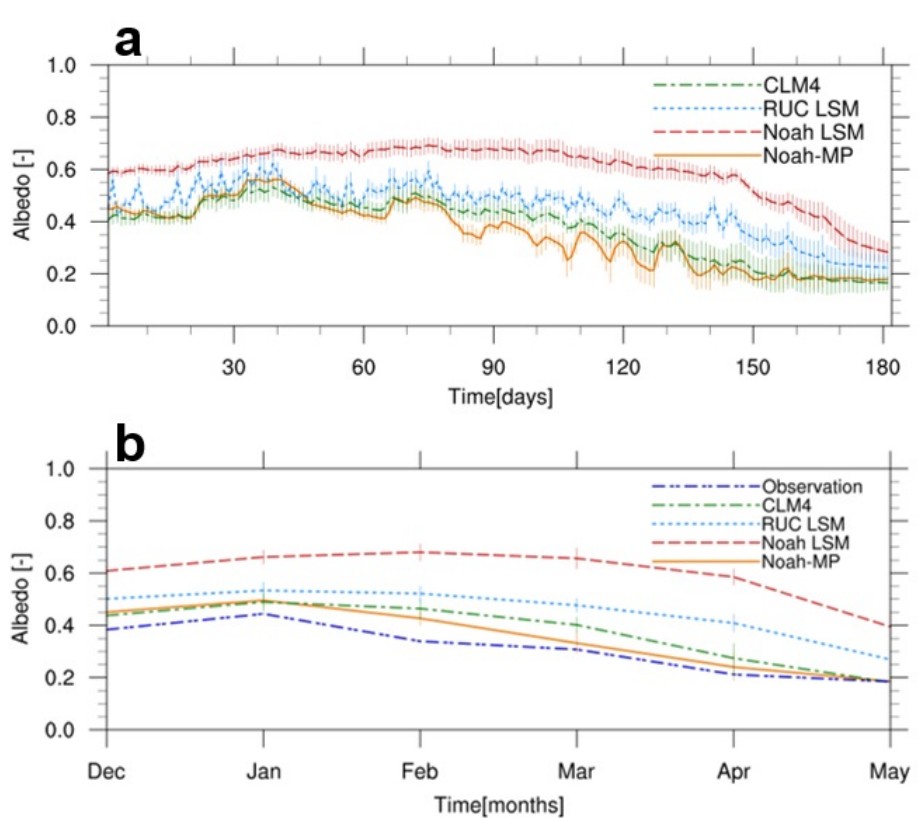

**Figure 13.** Same as in Fig. 9 but for surface albedo.





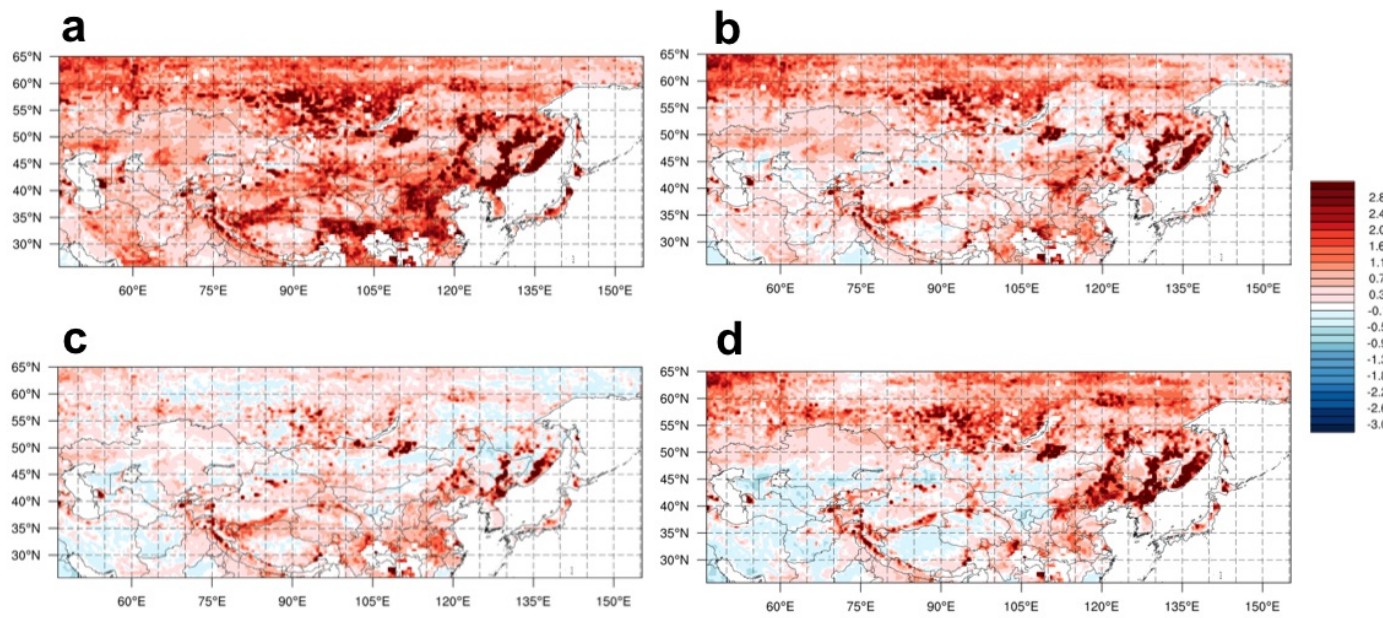

**Figure 14.** Same as in Fig. 2 but for the relative bias of surface albedo.

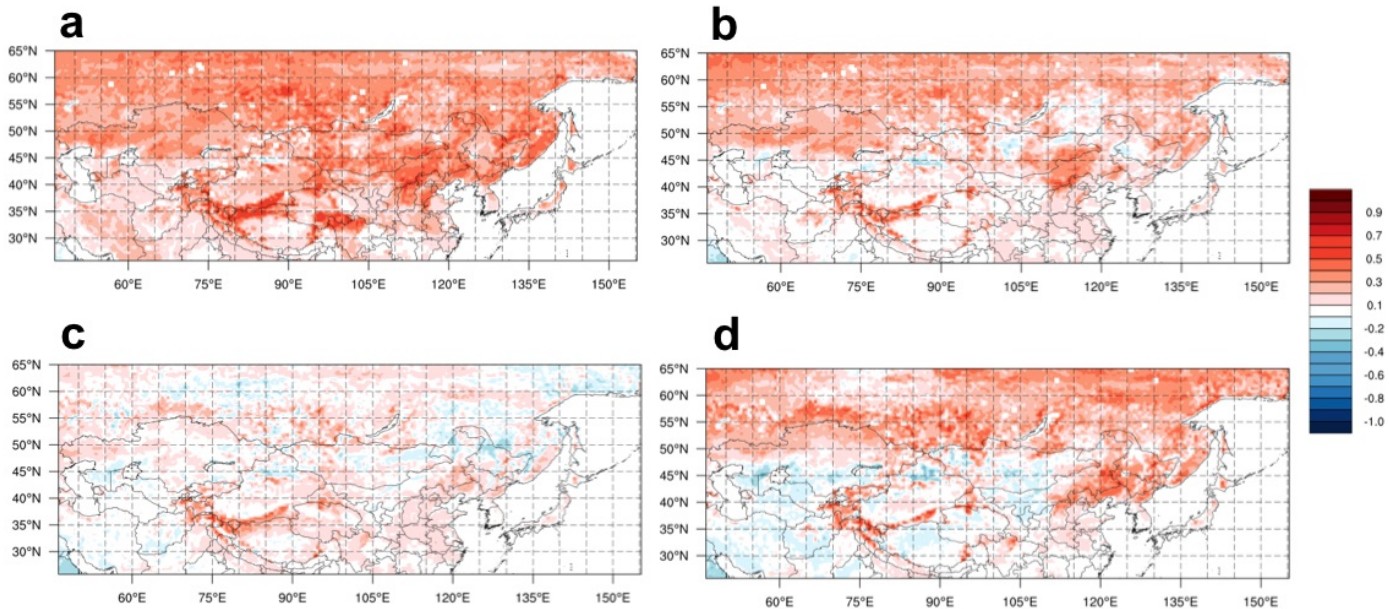

**Figure 15.** Same as in Fig. 2 but for the mean difference of surface albedo.





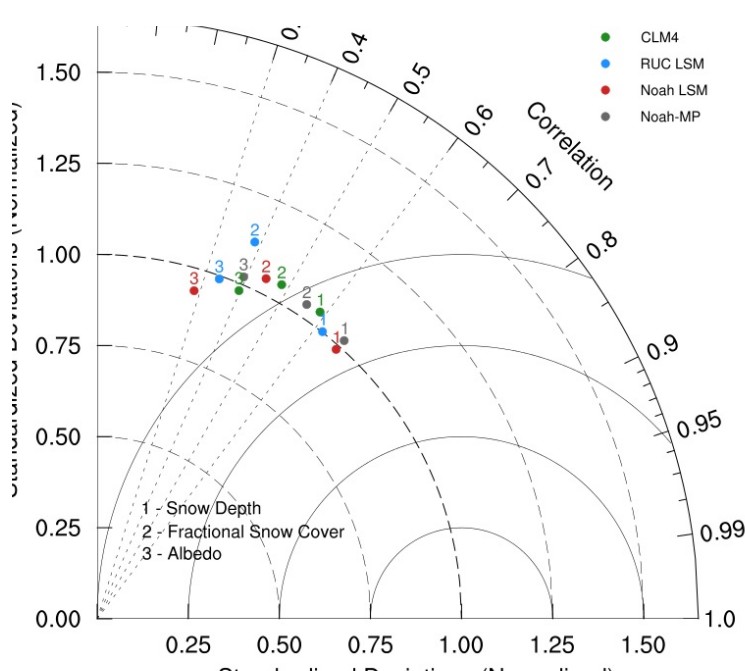

**Figure 16.** A Taylor diagram for all three variables and four LSMs. The LSMs are represented in the colored dots while the variables of corresponding LSM are represented in the colored numbers.