# Peer review of "Uncertainty in predicting the Eurasian snow: Intercomparison of land surface models coupled to a regional climate model"

_The Cryosphere, 2019_

## Referee Comment (RC1) · Anonymous Referee #1 · 12 Mar 2019

Remarks to the Authors

Review of "Uncertainty in predicting the Eurasian snow: Intercomparison of land surface models coupled to a regional climate model" by Da-Eun Kim and Seon Ki Park

The Cryosphere Discuss. Manuscript Number: tc-2019-15
* * *
General comments:

In this study, the authors performed inter-comparison of the performance of four land surface models (LSMs) simulating snow depth, fractional snow cover, and albedo in

[Figure]

Eurasia. These four LSMs are the Unified Noah land-surface model (Noah LSM), the Noah LSM with multi-parameterization options (Noah-MP), the Rapid Update Cycle (RUC), and the Community Land Model version 4 (CLM4). All the LSMs were coupled with the Weather Research and Forecasting (WRF) atmospheric model, then the inter-comparison was performed during the period from 1 June 2009 to 31 August 2010, which includes a model spin-up period of 6 month. From model validation results, the authors argue that the performance of Noah-MP was the best among these four LSMs in terms of reproducing measured snow depth, fractional snow cover, and albedo during the study period. Then, they highlight that simulation of the Eurasian snow cover is strongly affected by the choice of LSMs coupled with regional/global atmospheric models.

After reading this manuscript, I had several concerns about their study as follows. Please note that page and line numbers are denoted by "P" and "L", respectively.

- The main conclusion of this study "prediction of the Eurasian snow cover is sensitive to the choice of LSMs coupled to the global/regional climate models, and hence the future climate projections" (P. 1, L. 14 ∼ 15) is a matter of course. In the book by Armstrong and Brun (2008), it is presented clearly that simulation results by a LSM can change dramatically depending on the choice of model setting. In case several LSMs are compared, the difference can become much larger as reported by Etchevers (2004) and Krinner et al. (2018). Therefore, this reviewer thinks that there is nothing new in the present study.

- It is true that a model inter-comparison study is sometimes very useful and informative; however, this reviewer thinks that its purposes, protocol, and analysis strategy should be examined carefully in advance as performed by Krinner et al. (2018). It is well known that each LSM have their own purposes. In general, requirement level from a parent atmospheric model to a LSM is higher if a coupled system that consists of the atmospheric model and the LSM is used for long-term climate simulations, whereas, the requirement level would be relatively low if the coupled system is used for shortterm weather forecasts. It is because temporal evolution of snow physical conditions are much slower than that of atmospheric physical conditions. These imply that even a very simple LSM is sometimes very useful for a weather forecast model as long as the LSM can give reasonable bottom boundary conditions of the atmosphere. In addition, there is a possibility that some LSMs were tuned to give best performance in a specific area. I do not know in detail about WRF as well as the LSMs used here; however, I suspect some (physically simple) LSMs used here (for example, Noah LSM) should be used only for weather forecasts, although relatively detailed LSMs like CLM4 is suitable to be used for both purposes; the present inter-comparison procedure is a bit unfair for some simple LSMs (regarding the problem of the inter-comparison procedure, I mention below). Therefore, this reviewer was not convinced fully that performing this kind of inter-comparison is necessary and important.

- The inter-comparison procedure used here is too simple to rely on: The authors present mean difference, relative bias, root mean square error, and spatial correlation coefficient (Table 3); however, they are spatially and temporally averaged (in the model domain shown in Fig. 1 as well as the study period) (P. 10, L. 28 ∼ 29). In addition, their "observation data", which are the Canadian Meteorological Centre (CMC) Daily Snow Depth Analysis Data and the Moderate Resolution Imaging Spectroradiometer (MODIS) albedo, are not direct measurements; they can have error, which are generally larger than the error involved in direct measurements. This reviewer strongly recommends performing thorough model evaluation at the sites, where detailed in-situ meteorological and snow measurements data are available. Figures 6, 7, 10, 11, 14, and 15 suggest that model performance and characteristics change from place to place dramatically. The authors suggest considerable effects of canopy on reproducing realistic snow depths (P. 10, L. 5 ∼ 8). Maybe, assembling and summarizing point validation results depending on land surface types is valuable and informative for readers. Furthermore, detailed model validations in terms of e.g., precipitation, downward radiations, surface atmospheric temperature, humidity, wind speed, pressure, snow depth, albedo, land surface temperature, etc are certainly needed to understand "why different LSM produces different results in snow-related parameters" (P. 2, L. 30 ∼ 31). Due to the lack of detailed model validation results, I am not sure whether their suggestions regarding the further model improvement (P. 15, L. 15 ∼ 21) are reliable or not. Substantial efforts are needed to increase the reliability of their arguments.

- Finally, I have to suggest that using so many similar figures (e.g., Figs. 2, 7, 8, 10, 11, 12, 14, and 15) give the impression that the manuscript is a bit verbose.

Overall, this reviewer thinks that the current version of the manuscript cannot be published in the journal The Cryosphere.

––––––––––––––––––––––––––––––––––––––––––––––––––––––––––––––––––

Specific comments (major)

P. 1, L. 12∼14: As far as I learned from this manuscript, CLM4 is the most detailed and sophisticated model among the four LSMs in terms of incorporated snow physics. For example, CLM4 is the only model that can calculate snow albedo considering the effects of snow grain growth and light-absorbing impurities explicitly. However, the authors report that Noah-MP's performance was the best during the study period. I think there is a possibility that atmospheric conditions simulated by WRF was not realistic, which induced some compensations in the coupled system. Or, there is another possibility that given mass concentrations of light-absorbing impurities in CLM4 are not realistic as mentioned below. Therefore, I strongly recommend again to conduct detailed model validations in terms of e.g., precipitation, downward radiations, surface atmospheric temperature, humidity, wind speed, pressure, snow depth, albedo, land surface temperature, etc as mentioned above.

P. 4, L. 1: The authors state that Ms in equations (1) and (2) is snowmelt rate. Does it mean that meltwater produced in the model as well as liquid precipitation (rainfall) runoff instantaneously? No refreezing process in the model? If so, describe it, then state that Ms cannot be negative.

P. 6, L. 12 ∼ 13: How did the authors give mass concentrations of black carbon, dust, and organic carbon for CLM4? Giving realistic values for these light-absorbing impurities is crucial for accurate and reliable model inter-comparison.

P. 9, L. 14 ∼ 30: Before discussing features of calculated snow depths (even qualitatively), validations of simulated snow depths are needed. Many readers would not understand why the authors argue calculated snow depth patterns are "reasonable" (L. 15).

Figure 6: According to Table 3, CLM4 tended to underestimate snow depth; however, Fig. 6d indicates that CLM4 overestimates snow depth like other LSMs. Please discuss.

––––––––––––––––––––––––––––––––––––––––––––––––––––––––––––––

Specific comments (minor)

P. 1, L. 11 ∼ 12: It is unnecessary to mention "Bidirectional Reflectance Distribution Function (BRDF)" here. BRDF data are not used to validate LSMs.

P. 1, L. 21 ∼ 22: I think the sentence "in winter the land surface temperature is higher than the air temperature, whereas in spring the former becomes lower than the latter since snow reflects solar radiation with its high albedo (e.g., Park et al., 2017)." is unnecessary, because this situation is not always true in all over the cryosphere.

P. 1, L. 22 ∼ P. 2, L. 1: References for the statement "The temperature gradient between the land surface and the ocean is one of the important factors that influences the atmospheric circulations." are needed.

P. 2, L. 5: Describe quantitatively.

P. 2, L. 8 ∼ 10: The authors can merge these two sentences "Furthermore, the Eurasian snow is highly related with the climate and weather systems in Asia. Many studies have been conducted on the correlation between the South Asian climate/weather systems and the Eurasian snow.".

P. 2, L. 11: Explain the interaction mechanism mentioned here briefly.

P. 2, L. 14 ∼ 16: What is the key findings of these studies on the "interactions"?

P. 2, L. 30: The 1st goal sounds strange. The developers as well as most users of these LSMs might already understand them.

Section 2.1: Consider to add a table describing key differences in snow physical processes incorporated in the LSMs

P. 4, L. 1 ∼ 2: Indicate directions of energy fluxes mentioned here.

P. 4, L. 11: "tuning parameter" of what?

P. 4, L. 30: What do the authors mean by "destructive metamorphism"?

P. 4, L. 33: "Melting factor" might be a tuning parameter. What effects can we expect from the modulation of this parameter?

P. 5: L. 6 ∼ 8: The explanation on "snow age" is difficult to understand. Why do the authors relate snow age to grain growth effect and soot effect?

P. 6, L. 3: Change 2.5 K to 275.65 K

P. 6, L. 15: How often did the authors perform initialization of the atmosphere and land? Only at the beginning of the calculation period (1 June 2009) (climate simulation mode) or every day (weather forecast mode)?

P. 6, L. 25: It is not necessary to mention vorticity and ozone here.

P. 7, L. 3: What is "snow-climate classes"? Please explain more.

P. 7, L. 11: What does the "differences" tell us? Please describe.

P. 7, L. 30: The intention of "microphysical options" is not clear. Please detail more.

[Figure]
* * *
Technical corrections:

P. 1, L. 24: "temperature gradient" -> "horizontal temperature gradient"

P. 2, L. 19: "soil and vegetations" -> "soil, snow, and vegetation"

P. 6, L. 13: "ice effective grain size" -> "effective snow grain size"

P. 6, L. 15: "boundary conditions" -> "boundary conditions of the atmosphere"

P. 7, L. 10: "very high reflections" -> "relatively high reflections"

P. 7, L. 10: "very low reflections" -> "relatively low reflections"

P. 8, L. 1: Change "fractional snow cover with —" to "fractional snow cover Fs with —", then, remove "Fs is fractional snow cover —" in L. 3.

P. 8, L. 4 ∼ 6: Move "Among the LSMs coupled to WRF, we excluded the Thermal Diffusion (TD) scheme (Dudhia, 1996) and the Pleim-Xiu LSM (Pleim and Xiu, 1995), since they do not consider the snow depth in their snow scheme." to Sect. 2.1.

P. 9, L. 1: "value" -> "value of snow depth"

Table 1: Cite Oleson et al. (2010) for CLM4.

Table 3: Indicate units.

Figures 1a and 2: Consider to arrange easy-to-understand numbers in the color bars.

Figures 3, 4, 5, 9, and 13: The x-axis should not be noted in the number of days after 1 December 2009. Using date is much better to understand.

Figures 4b and 4c: The unit of y-axis is strange?

Figure 4 caption: "The amount of (a) —" -> "The area-averaged amounts of (a) —"

Figure 5 caption: Why not 273.15 K rather than 273.16 K?

—————————————————————————————————————————————————

References:

Armstrong, R. L. and Brun, E. (Eds.): Snow and Climate: Physical Processes, Surface Energy Exchange and Modeling, Cambridge Univ. Press, Cambridge, UK, 2008.

Etchevers, P., et al.: Validation of the energy budget of an alpine snowpack simulated by several snow models (SnowMIP project), Ann. Glaciol., 38, 150–158, doi:10.3189/172756404781814825, 2004.

Krinner et al.: ESM-SnowMIP: assessing snow models and quantifying snow-related climate feedbacks, Geosci. Model Dev., 11, 5027-5049, doi:10.5194/gmd-11-5027-2018, 2018.

---

## Referee Comment (RC2) · Anonymous Referee #2 · 15 Mar 2019

In this study, the authors tried to address the uncertainty in predicting the Eurasian snow by inter-comparing the performances of four land surface models coupled to WRF. They have four goals to achieve but the purpose of the work is not clearly articulated. The manuscript can be accepted to be published only after some major concerns have been addressed:

(1)Many studies show errors in the input and validation data, rather than model formulation, seem to be the greatest factor affecting model performance. The manuscript lacks of discussion of the quality for those "observation" used to evaluate the model performance.

[Figure]

(2) Among four goals of this paper only the third one has been addressed sufficiently. The first two haven't been explained clearly. The reader can easily understand the difference between four models if a table is used to show the different treatment of snow albedo, snow density, snow compaction, snow interception, snow age, and etc. The authors need to make some sensitivity tests like the different microphysics schemes to address and make any conclusion on the last goal of this paper.

(3)For fair inter-comparison the forcing should be as close as possible. For snow predictions how to differenciate snowfall or rainfall from the total precipitation is critical. This process should be mainly determined by microphysics scheme. The authors should use the same MP for all models to determine the amount of snowfall and rainfall rather than the empirical method by each model.

(4) For the most part, snow models are built on similar principles. The greatest differences are found in how each model parameterizes individual processes (e.g., surface albedo and snow compaction). Parameterization choices naturally span a wide range of complexities. Ensemble is a promising way to reduce these uncertainties. It would be greatly beneficial to the title of this paper if the authors can also compare the performance of the ensemble mean of four models.

(5)Noah MP itself has so many options to choose. Many of them can have significant impact on the snow prediction. The authors should have optimal options before comparing it to the other three models.

(6)The quality of figure need to be improved. Some of them are very blurry and hard to read. Like figure 4a only 6 curves can be seen.

---

## Author Comment (AC1) · 8 Apr 2019

**Reply to the Comments by Referee #1 for Manuscript tc-2019-15**

**General comments:** *In this study, the authors performed inter-comparison of the performance of four land surface models (LSMs) simulating snow depth, fractional snow cover, and albedo in Eurasia. These four LSMs are the Unified Noah land-surface model (Noah LSM), the Noah LSM with multi-parameterization options (Noah-MP), the Rapid Update Cycle (RUC), and the Community Land Model version 4 (CLM4). All the LSMs were coupled with the Weather Research and Forecasting (WRF) atmospheric model, then the inter-comparison was performed during the period from 1 June 2009 to 31 August 2010, which includes a model spin-up period of 6 month. From model validation results, the authors argue that the performance of Noah-MP was the best among these four LSMs in terms of reproducing measured snow depth, fractional snow cover, and albedo during the study period. Then, they highlight that simulation of the Eurasian snow cover is strongly affected by the choice of LSMs coupled with regional/global atmospheric models.*

⇒ The authors appreciate the careful and valuable comments by the referee on this study, which enabled us to improve the quality of the manuscript significantly. We have made our best effort to revise the manuscript based on the referee's comments and suggestions. In the following, we made an item-by-item response to the specific comments by the referee.

*After reading this manuscript, I had several concerns about their study as follows. Please note that page and line numbers are denoted by "P" and "L", respectively.*

- *The main conclusion of this study "prediction of the Eurasian snow cover is sensitive to the choice of LSMs coupled to the global/regional climate models, and hence the future climate projections" (P. 1, L. 14 ∼ 15) is a matter of course. In the book by Armstrong and Brun (2008), it is presented clearly that simulation results by a LSM can change dramatically depending on the choice of model setting. In case several LSMs are compared, the difference can become much larger as reported by Etchevers (2004) and Krinner et al. (2018). Therefore, this reviewer thinks that there is nothing new in the present study.*

  ⇒ We appreciate this comment by the referee, but we believe that our research has its own value and unique/new approaches and findings compared to those studies mentioned by the referee. After reading the two papers, i.e., Etchevers et al. (2004) and Krinner et al. (2018), and having serious consideration on the uniqueness of our study, we could not fully agree to the referee's statement that our study has nothing new, compared to those studies. To avoid any confusion or misunderstanding, we have tried to address the uniqueness of our study more clearly in the revised manuscript, based on the following speculation:

  1. It is generally well known that the simulation results (e.g., energy/water budget components) can be different for different LSMs and even for different setting in a selected LSM, even before those studies that the referee mentioned (e.g., Henderson-Sellers et al., 1993, 1995, 1996; Pitman and Henderson-Sellers, 1998). However, this general knowledge does not mean that further studies are not necessary: a bunch of studies have been published since then for intercomparison of LSMs, with similar conclusions, including those mentioned by the referee. Although a statement of our

conclusions may be similar to the conclusions from the previous studies, we believe that the specific contents of our study are very unique.

2. It is true that simulation results by an LSM can change dramatically depending on the choice of model setting, especially at the top surface layer that interacts directly with the atmosphere. The unique structure and scheme of each LSM has a large impact on calculation of soil heat flux, soil moisture contents, soil temperature, etc., by complex interaction through the multiple soil/snow layers with initial forcings from atmosphere (e.g., Liang et al., 1999). However, most model intercomparison studies, including those referred to by the referee, had not discussed the characteristics of the snow-related physical processes of each model employed for the studies. We have uniquely tried **to understand the differences in physical processes of calculating snow-related processes in each model** that was employed in our study.

3. We acknowledge the importance of the model intercomparison studies by Etchevers et al. (2004) and Krinner et al. (2018). As these studies are through international collaborations, they include many models and observations based on abundant resources available from multiple institutions involved in the project. Although we cannot compete with them, using only limited resources, and employed only four LSMs for our study, we certainly have unique points that are different from their studies as described below:

   (a) Etchevers et al. (2004) made intercomparison among several snow models in validating the energy budgets but only in the **offline mode**; however, we made intercomparison of LSMs in the **online mode**, that is, **coupled to the atmospheric model**, allowing the feedbacks between the land surface and atmosphere. Furthermore, among the models they have employed, **only Noah LSM is in common** with our study. In other words, they did not include three other models that were used in our study. In addition, **they neither discussed the differences in computing the snow-related physical processes of each model nor compared the results in terms of this viewpoint**.

   (b) Krinner et al. (2018) provided a detailed description of a project that they plan to do with some preliminary results. Their figures showed the comparisons between some site measurements and multi-model ensemble results in terms of snow-related parameters such as albedo, snow water equivalent, snow depth, etc. Although they showed the ensemble results, **they lack discussions on the performance of each LSM**. They described the plans of global simulations with land-atmosphere coupling or land only options but **no results were shown yet**. Their plans are very good and exciting as an international collaborative effort, and we have no doubt that their project will give another important contribution to our community; however, they have no results yet comparable to ours and only discussed the "expected outcome" and "possible actions" (see their section 4). Even when this project is completed successfully, our results are unique compared with theirs because **we performed the regional coupled simulations focusing on Eurasia** while **they will perform the global simulations**. Furthermore, among many models in their planned project, **only RUC is in common** with our study and again **they do not currently include discussions on the snow-related physical processes** used in their models.

4. We also address the uniqueness of our study in terms of **the study domain – Eurasia**. Many previous studies have discussed the relationship between the Eurasian snow and regional/global climate variation (e.g., to mention just a few, Bernett et al., 1989; Bamzai and Shukla, 1999; Cohen and Entekhabi, 1999; Kripalani et al., 2002; Liu and Yanai, 2002; Wu and Kirtman, 2007; Wu et al., 2009; Allen and Zender, 2011), demonstrating the significant effect of Eurasian snow on the interaction between land surface and atmosphere. Therefore, it is worthwhile to perform the intercomparison study in order to examine how different physical processes in the LSMs produce different behaviors in snow-related parameters, especially **over the region that has strong interactions/feedbacks between the snow-covered land surface and atmosphere**.

5. Since each LSM has its own physical/dynamical characteristics, the model results can vary depending on target surface variables, region, and simulating period. As a simple example, in our experiments, the Noah-MP showed the best performance in predicting snow-related variables over Eurasia. On the other hand, the RUC LSM and CLM4 represented better results in predicting the 2 m temperature during summer, especially over the area between 90–120°E (see Fig. R1 below). Furthermore, the Noah LSM also showed better results in the 2 m temperature than Noah-MP. In Fig. R1, the domain-averaged values were −2.67 for CLM4, −2.46 for RUC, −3.41 for Noah-MP, and −3.07 for Noah LSM.

    Jin et al. (2010) also conducted a sensitivity test with four land surface models over the western United States and concluded that the CLM3, the predecessor of CLM4, was the best in predicting snow water equivalent and surface temperature. Thus it is important **to understand the differences in the model behviors based on the differences in the model physical processes** rather than to merely report the different results from various models.

[Figure]

**Fig. R1.** The mean difference of 2 m temperature in summer for (a) the Noah LSM, (b) the RUC LSM, (c) the Noah-MP, and (d) the CLM4.

- *It is true that a model inter-comparison study is sometimes very useful and informative; however, this reviewer thinks that its purposes, protocol, and analysis strategy should be examined*

*carefully in advance as performed by Krinner et al. (2018). It is well known that each LSM have their own purposes. In general, requirement level from a parent atmospheric model to a LSM is higher if a coupled system that consists of the atmospheric model and the LSM is used for long-term climate simulations, whereas, the requirement level would be relatively low if the coupled system is used for short-term weather forecasts. It is because temporal evolution of snow physical conditions are much slower than that of atmospheric physical conditions. These imply that even a very simple LSM is sometimes very useful for a weather forecast model as long as the LSM can give reasonable bottom boundary conditions of the atmosphere. In addition, there is a possibility that some LSMs were tuned to give best performance in a specific area. I do not know in detail about WRF as well as the LSMs used here; however, I suspect some (physically simple) LSMs used here (for example, Noah LSM) should be used only for weather forecasts, although relatively detailed LSMs like CLM4 is suitable to be used for both purposes; the present inter-comparison procedure, I mention below). Therefore, this reviewer was not convinced fully that performing this kind of inter-comparison is necessary and important.*

⇒ We agree to the referee's comment that a model intercomparison study needs to be examined in advance in terms of its purpose, protocol, and analysis strategy, as performed by Krinner et al. (2018). Such a paper is based on a national/international project, whose plan draws big attention from the scientific community; thus a report describing the project's goal, strategy and plans is mostly accepted for publication in major journals. However, our study is not based on such a national/international project but just out of a laboratory-level thesis research, whose research plan is usually not accepted for publication. For this kind of individual laboratory-level research, the referee's request to publish the protocol and strategy in advance seems to carry too far. Although we could not publish the purpose, protocol and analysis strategy of our planned experiments in advance of this study, we have done our best to reflect such things in the revised manuscript. Nevertheless, we still believe that performing this kind of intercomparison is necessary and important, though it is not based on a national/international project, for the following reasons:

1. It is true that even a very simple LSM is sometimes very useful for a weather forecast model as long as the LSM can give reasonable bottom boundary conditions of the atmosphere. However, it does not necessarily mean that more complex LSMs are better for a weather forecast model: no matter how complex the LSM is, short term forecasts cannot stabilize LSM over all soil/snow layers, and the surface variables, which can have sufficient impact on the atmosphere even during the short period, can be predicted even without subsurface physics. Sometimes, accuracy of the input background field can be more important factor.

2. Simple LSMs can also be coupled with regional/global atmospheric model to predict both weather and climate. For example, Noah LSM, which is considered as a simple LSM, is one of the most widely used LSMs in weather and climate simulation studies. It is the main core of the Global Land Data Assimilation System (GLDAS; Rodell et al., 2004), which had been used for one of the base data for model evaluation in the Coupled Model Intercomparison Project Phase 5 (CMIP5; e.g., Sheffield et al., 2013; Harding et al., 2013; Yuan and Quiring, 2017; Sippel et al., 2017). The Noah LSM is also used in the model intercomparison study for simulating evapotranspiration (Ukkola et al., 2016) and snow aspects (Etchevers et al., 2004). The National Center for Environmental Prediction (NCEP) weather/climate prediction models

employ Noah LSM for both operational and research applications, and for data assimilation as well. Additionally, the Noah LSM is also adopted and coupled to a new global model being developed at the Korea Institute of Atmospheric Prediction Systems (KIAPS; Koo et al., 2017), which will be used for both weather and climate predictions.

3. The WRF model, used in this study, has recently been recognized and used widely as a regional climate model in many regional climate simulation and/or projection problems (e.g., to mention just a few, Bukovsky and Karoly, 2009; Chotamonsak et al., 2011; Zhang et al., 2012; Dasari et al., 2014; Abdallah et al., 2015; Alsarraf and van den Broeke, 2015; Oaida et al., 2015; Ramarohetra et al., 2015; Raghavan et al., 2016; Ratna et al., 2017; Mooney et al., 2019). Therefore, all the LSM options in WRF basically can be used for regional climate problems; however, the performance of each LSM needs to be evaluated before it is used for climate applications, as in this study.

4. In summary, considering that all of the LSMs simulated in our study are used for both weather and climate predictions, we believe that this kind of intercomparison is essential and important. Furthermore, as the WRF model is one of the most widely used models for researches in both weather and climate problems, we believe that the findings in this study are valuable to the researchers who are interested in accurate prediction of the Eurasian snow.

- *The inter-comparison procedure used here is too simple to rely on: The authors present mean difference, relative bias, root mean square error, and spatial correlation coefficient (Table 3); however, they are spatially and temporally averaged (in the model domain shown in Fig. 1 as well as the study period) (P. 10, L. 28 ∼ 29). In addition, their "observation data", which are the Canadian Meteorological Centre (CMC) Daily Snow Depth Analysis Data and the Moderate Resolution Imaging Spectroradiometer (MODIS) albedo, are not direct measurements; they can have error, which are generally larger than the error involved in direct measurements. This reviewer strongly recommends performing thorough model evaluation at the sites, where detailed in-situ meteorological and snow measurements data are available. Figures 6, 7, 10, 11, 14, and 15 suggest that model performance and characteristics change from place to place dramatically. The authors suggest considerable effects of canopy on reproducing realistic snow depths (P. 10, L. 5 ∼ 8). Maybe, assembling and summarizing point validation results depending on land surface types is valuable and informative for readers. Furthermore, detailed model validations in terms of e.g., precipitation, downward radiations, surface atmospheric temperature, humidity, wind speed, pressure, snow depth, albedo, land surface temperature, etc., are certainly needed to understand "why different LSM produces different results in snow-related parameters" (P.2, L. 30 ∼ 31). Due to the lack of detailed model validation results, I am not sure whether their suggestions regarding the further model improvement (P. 15, L. 15 ∼ 21) are reliable or not. Substantial efforts are needed to increase the reliability of their arguments.*

  ⇒ In principle, we agree to the referee's comment on the need of comprehensive validation of the employed models. However, we believe that the validation process as well as the observation data used in our study is sufficiently reliable by following reasons:

    1. The referee suggested a detailed model validations against almost all related variables, e.g., precipitation, downward radiations, surface atmospheric temperature, humidity, wind speed, pressure, snow depth, albedo, land surface temperature, and so on, with detailed in-situ meteorological and snow measurement data. It is very

desirable to perform such a comprehensive validation but in many real situation it is neither feasible nor effective. As far as we know, none of the model intercomparison study performed such a comprehensive validation. For example, Etchevers et al. (2004) validated the models against only the snow energy budget components, such as net short- and longwave radiation, latent/sensible heat fluxes and snowmelt. Krinner et al. (2018) validated the models in terms of only four parameters – snow water equivalent, albedo, snow depth, and soil moisture – only in ensemble spread sense.

2. We believe that the model validation should be preformed appropriately to suffice the goal of a given research. The comprehensive validation against detailed in situ measurements, as the referee suggested, is usually done at the stage of developing and/or improving a model. Such a validation test is mostly conducted using a one-dimensional version of the model at the site or at a grid point very close to the site where the measurements are made. As the purpose of our study is to examine the performance of LSMs in WRF in predicting the "Eurasian snow", we have validated the model in terms of overall performance over a wide area rather than at a point or a specific location for the following reasons:

   (a) Although it would be really fantastic to do a comprehensive validation, as an individual research group, it is not feasible to collect the in situ measurement data that sufficiently and reasonably cover the whole Eurasia region. Assembling and summarizing the point validation results is an excellent idea but it is practically almost impossible for us to collect such data that represent various land surface types in the whole Eurasia region. It requires multi-countries international collaborations but our study is not in that category. Furthermore, we are not aiming at developing the model or improving the parameterization schemes; thus not requiring such a comprehensive model validation using the detailed point measurements.

   (b) We could have used some in situ data, collected in South Korea, but most of the measurements were made far from the model grid points: they require interpolations for validation which result in additional errors. We also think that, for this kind of study, it is meaningless to make validation at only a couple of sites in South Korea rather than over the whole study domain. Therefore, we decided to use the satellite data which already have been validated and processed through many quality checks from the previous studies.

3. The Canadian Meteorological Centre (CMC) snow depth data and MODIS/Terra Snow Cover Monthly L3 Global 0.05Deg CMG, Version 6 are the most widely used data in many model evaluation studies, being considered as the truth values (e.g., Drusch et al., 2004; Rodell and Houser, 2004; Niu and Yang 2007; Parajka and Bloschl, 2008; Hall et al., 2010; Reichle et al., 2011; Yang et al., 2011; Nester et al., 2012; Hancock et al., 2013; Liu et al., 2013; Kumar et al., 2014, 2015; Zhou et al., 2014; Dawson et al., 2016; Toure et al., 2016). Therefore, many previous studies have already validated these observation datasets against in situ data and demonstrated the reliability of them (e.g., Hall and Riggs, 2007; Liu et al., 2009; Tao et al., 2015; Cooper et al., 2018). Furthermore, Chervenkov et al. (2015) addressed the usefulness of satellite data in validation by stating that

   *"Satellite earth snow observation products have the needed spatial and temporal consistency, which allows comparisons with model output over continuous*

*area and time frames. The absence of this consistency of the point measurements is an inherent weakness of every statistical evaluation procedure based upon them and thus utilizing satellite data is a significant step ahead in the quantitative snow cover assessment."*

Based on all those previous studies, we do not see any problem in using the CMC data and MODIS satellite data for the model validations in our study.

4. We performed additional validation after reading the referee's comment. Firstly, we tried to see the bias and the RMSE for different land categories. In Fig. R2, as all four models represent similar patterns, we just compare two LSMs – the Noah LSM that shows the lowest accuracy (Figs. R2 a and c) and the Noah-MP that represents the highest accuracy (Figs. R2 b and d). We also did the analysis on the daily variation of bias and RMSE through the experiment period, but the results show almost the same values throughout the period. With these results, as we already mentioned in our study, the highest RMSE appeared over the grassland by highly underestimating the snow depth.

[Figure]

**Fig. R2.** The bias for (a) the Noah LSM, (b) the Noah-MP and the RMSE for (c) the Noah LSM, (d) the Noah-MP in simulating snow depth. The number and each color represent the categories which are represented in Fig. 1b in our original manuscript.

5. We also have added the ERA 5 data for further validation. Figure R3 shows snow depth, surface temperature, and wind speed for all the four models, validating with the CMC and ERA5 datasets. For the snow depth, we added the ERA5 data to the original graph in our study. The overall increasing and decreasing rate pattern of ERA5 is similar to that of the CMC data throughout the period but represents a lower snow depth.

In the original manuscript, we did not validate the models against meteorological data. As the ERA5 data also include meteorological data, we have performed additional validation in terms of surface temperature and wind speed. Figure R3b depict the validation of the models against the ERA5 surface temperature data. All the models tend to predict the surface temperature lower than ERA5 during winter, whereas this trend reverses at the end of the spring. The cold bias of surface temperature during winter seems to be related to an overestimation of snow depth by reducing the snow melting effect. Reversely, during spring, the snow melting effect

is overestimated by the models, causing a steeper increasing rate of temperature in the models than in ERA5.

Lastly, we have conducted the model validation against the ERA5 wind speed data (Fig. R3c). All the models tend to overestimate wind speed during the whole period; however, they represent variation patterns similar to ERA5. Note that we have found significant bias and RMSE over the grassland in the snow depth simulation (see Fig. R2). Considering that the grassland is highly affected by the wind, as we also mentioned in the original manuscript, the overestimation of wind speed in all the models can bring about a negative bias in predicting the snow depth.

[Figure]

**Fig. R3.** Validation of the models in terms of (a) snow depth, (b) surface temperature, and (c) wind speed during winter and spring over the study domain. Both the CMC and ERA5 data were used for (a), whereas only the ERA5 data were used for (b) and (c).

6. In regard of the validation method, the statistical parameters employed in this study such as mean difference, relative bias, root mean square error, and spatial correlation coefficient have been widely used for validation purpose (see, e.g., Jacob et al., 2007; Abramowitz et al., 2008; Walsh et al., 2008; Choi et al., 2010; Gilliam and Pleim, 2010). Therefore, we believe that those statistical parameters have adequately validated the model performance both in time and space.

- *Finally, I have to suggest that using so many similar figures (e.g., Figs. 2, 7, 8, 10, 11, 12, 14, 15) give the impression that the manuscript is a bit verbose.*

  ⇒ We have sorted out the figures in the revised manuscript to avoid verbosity.

**Specific comments (major):**

1. *P. 1, L. 12 ∼ 14: As far as I learned from this manuscript, CLM4 is the most detailed and sophisticated model among the four LSMs in terms of incorporated snow physics. For example, CLM4 is the only model that can calculate snow albedo considering the effects of snow grain growth and light-absorbing impurities explicitly. However, the authors report that Noah-MP's performance was the best during the study period. I think there is a possibility that atmospheric conditions simulated by WRF was not realistic, which induced some compensations in the coupled system. Or, there is another possibility that given mass concentrations of light-absorbing impurities in CLM4 are not realistic as mentioned below. Therefore, I strongly recommend again to conduct detailed model validations in terms of e.g., precipitation, downward radiations, surface atmospheric temperature, humidity, wind speed, pressure, snow depth, albedo, land surface temperature, etc. as mentioned above.*

   ⇒ As the referee has pointed out, CLM4 is the most sophisticated model in this study. However, a more complex and sophisticated model does not necessarily produce better results, as demonstrated in our study and other previous studies. For example, Chervenkov et al. (2015) compared the performances of two LSMs – CLM3.5 and BATS – both coupled to a regional climate model (RegCM4), obtaining the initial and boundary conditions from either the ERA-Interim data or the NCEP/NCAR Reanalysis-2 data. Note that CLM3.5 was a more advanced package than and theoretically superior to BATS. They concluded that no model configuration, among the four (2 LSMs and 2 background data), performs significantly better than the others. In particular, for the snow water equivalent, CLM3.5 showed much larger bias than BATS, when the ERA-Interim data was used as the background. This implies that, no matter how sophisticated the physical processes are in an LSM, **its performance when coupled to the regional climate models is strongly dependent on the initial and boundary conditions** provided by the background data.

   Steiner et al. (2005) also compared the performance of CLM0 with BATS, both coupled with RegCM, using the ECMWF reanalysis data as the background. They found that BATS was in greater agreement with observations by producing a larger snow accumulation on the surface than CLM0. They attributed this to **a feedback mechanism through interactions in a coupled modeling system**: the reduced CLM0 snow cover caused a reduction in surface albedo and increased the amount of radiation absorbed, then in turn, increased the surface radiative energy fluxes, leading to higher winter temperatures in CLM0.

   We speculate that this is not a problem of validation and that we cannot obtain useful information even through the detailed and comprehensive validation with point measurements. In order to investigate the reason for the discrepancy of having worse results with more sophisticated (or better) model, it is much worthy to do many sensitivity experiments to see the effect of initial and lateral boundary conditions as well as

physical parameters, rather than to do detailed validations with a few point measurements; however, this kind of sensitivity experiments are out of the scope in this study. In addition, as the feedbacks through nonlinear interactions are too complex, it is not feasible to attack this kind of discrepancy problem and again it is out of accord with the goals and scopes in this study. Nonetheless we tried to make physical interpretation on this matter in the revised manuscript.

2. *P. 4, L. 1: The authors state that $M_s$ in equations (1) and (2) is snowmelt rate. Does it mean that meltwater produced in the model as well as liquid precipitation (rainfall) runoff instantaneously? No refreezing process in the model? If so, describe it, then state that $M_s$ cannot be negative.*

   ⇒ The Noah LSM also considers the snow refreezing process, and the liquid precipitation (rainfall) runoff is calculated after the snow melt rate is decided. We describe below how the Noah LSM considers the snow melting processes by referring to the code of Noah LSM to show why $M_s$ cannot be negative. We have clearly described that $M_s$ is not negative in the revised manuscript. Please note that the following step-by-step explanation is based on the code and the Noah LSM description.

   (a) When precipitation occurs, the Noah LSM calculates the heat flux from snow surface to newly accumulating precipitation by assuming that the temperature of the snowfall striking the ground is surface temperature (lowest model level air temperature).

   (b) Then the Noah LSM calculates an effective snow-ground surface temperature ($T_{sg}$), based on heat fluxes between the snow pack and the soil and net radiation.

   (c) Using $T_{sg}$ obtained in (b), the Noah LSM determines whether snow melt will occur. If $T_{sg}$ is lower or equal to the freezing temperature, **no snow melt will occur**. Reversely, if $T_{sg}$ is higher than the freezing temperature, **snow melt will occur** and the temperature will be revised considering the latent heat released from the melting processes.

   (d) Then it also considers the evaporation (sublimation) effect. When the amount of potential evaporation exceeds the total amount of snow depth, the Noah LSM sets the snowpack to 0, which means that snow melt also goes to 0. Reversely, if the amount of evaporation (sublimation) is less than the total amount of snow depth, it updates snow depth by considering the evaporation (sublimation) effect.

   (e) Snow melt is derived from the integrated snow melt reduction rate, which is calculated depending on snow cover. If snow melt exceeds the existing snow, snow depth becomes the upper limit for snow melt. Reversely, when the existing depth of snow is larger than the amount of snow melt, snow depth will be reduced by the melting effect.

   (f) Finally, the Noah LSM adds the amount of snow melt to precipitation.

   (g) Runoff/baseflow is later calculated at the subroutine SFLX which is described as: "*a sub-driver for "Noah-LSM" family of physics subroutines for a soil/vegetation/snowpack LSM to update soil moisture, soil ice, soil temperature, skin temperature, snowpack water content, snow depth, and all terms of the surface energy balance and surface water balance (excluding input atmospheric forcings of downward radiation and precipitation)*".

3. *P. 6, L. 12 ∼ 13: How did the authors give mass concentrations of black carbon, dust,*

*and organic carbon for CLM4? Giving realistic values for these light-absorbing impurities is crucial for accurate and reliable model inter-comparison.*

⇒ The CLM4 considers 8 particle species within snow layer: hydrophilic black carbon, hydrophobic carbon, hydrophilic organic carbon, hydrophobic organic carbon, and four species of mineral when the CLM4 gets aerosol deposition rate from atmospheric forcing. However, the current WRF version used in this study does not include a chemical module, thus no consideration is made for this effect. This issue is apparently very interesting, but it is out of the scope of this study.

4. *P. 9, L. 14 ∼ 30: Before discussing features of calculated snow depths (even qualitatively), validations of simulated snow depths are needed. Many readers would not understand why the authors argue calculated snow depth patterns are "reasonable" (L. 15)*

⇒ We have already performed validations of the simulated snow depths (see Fig. 3 in the original manuscript). We have used the expression "reasonable" (P. 9, L. 15) in a general sense to imply that all the models are in general agreement with the pattern of snow depth observation, as we mentioned in P. 9, L. 15-16: "in general, snow depths tend to be deeper over the regions of higher altitude and latitude".

5. *Figure 6: According to Table 3, CLM4 tended to underestimate snow depth; however, Fig. 6d indicates that CLM4 overestimates snow depth like other LSMs. Please discuss.*

⇒ The mean difference (MD) value in Table 3 represents a domain-averaged quantitative difference between the model and the observation value. Therefore, if the domain includes highly underestimated grid points, even in a small region, the domain-averaged MD value can be negative. Figure R4 below represents the period-averaged MD values over the study domain for CLM4. As we mentioned, very large negative MD values appeared along the mountain brinks (e.g., the Tibetan Plateau) while the remaining area mostly represented relatively small positive values. By taking a domain average, the MD value will be negative, as shown in Table 3. However, in Fig. 6d, a single dot represents the model and observation value on every single grid point: the result indicates that CLM4 overestimates snow depth over almost whole domain. In conclusion, the MD values in Table 3 shows validation in the viewpoint of quantitative accuracy while the dot representation in Fig. 6d shows the overall simulation tendency over the whole domain.

**Specific comments (minor):**

1. *P. 1, L. 11 ∼ 12: It is unnecessary to mention "Bidirectional Reflectance Distribution Function (BRDF)" here. BRDF data are not used to validate LSMs.*

⇒ The albedo data we used in our study is derived from BRDF for the land bands (1–7) as well as three broad bands (0.4–0.7, 0.7–3.0, and 0.4–3.0 micrometers). As it includes the major method of data processing, we want to keep it but with additional explanation why we used the term "**MODIS Bidirectional Reflectance Distribution Function (BRDF)/Albedo Product**".

2. *P. 1, L. 21 ∼ 22: I think the sentence "in winter the land surface temperature is higher than the air temperature, whereas in spring the former becomes lower than the latter since snow*

[Figure]

**Fig. R4.** Mean difference of snow depth (m) for CLM4, averaged for a period from December 2009 to May 2010.

*reflects solar radiation with its high albedo (e.g., Park et al., 2017)." is unnecessary, because this situation is not always true in all over the cryosphere.*

⇒ We agree with the referee that the statement is not always true. We should have expressed it in a general way, and we modified this part as the following:

"*Snow inhibits direct heat exchange between land surface and atmosphere:* **in general**, *the land surface temperature is higher than the air temperature in winter, whereas the former becomes lower than the latter in spring as snow reflects solar radiation with its high albedo (e.g., Park et al., 2017).*"

3. *P. 1, L. 22 ∼ P. 2, L. 1: References for the statement "The temperature gradient between the land surface and the ocean is one of the important factors that influences the atmospheric circulations" are needed.*

⇒ We appreciate the referee's suggestion, and we have added several references in the revised manuscript: Li and Yanai, 1996; Chou, 2003; Qi et al., 2008; Turrent and Cavazos, 2009; Sun et al., 2010; Wu et al., 2012; Kamae et al., 2014; Roxy et al., 2015 (see the **References** section in this reply).

4. *P. 2, L. 5: Describe quantitatively.*

⇒ We have modified this part following the referee's suggestion. Chen et al. (2016) showed the region where snow covered the land surface at least 8 weeks (∼60 days) in 75% of the years between 1982 and 2013 (see Fig. R5 below). Other studies showed that ∼98% of the seasonal snow is located in the Northern Hemisphere (NH), and ∼42% of the land in NH is covered by snow for a significant duration (Dingman, 2002; Toure et al., 2016).

5. *P. 2, L. 8 ∼ 10: The authors can merge these two sentences "Furthermore, the Eurasian snow is highly related with the climate and weather systems in Asia. Many studies have*

[Figure]

**Fig. R5.** The region where snow covered the land surface at least 8 weeks in 75% of the years between 1982 and 2013 over Northern Hemisphere (Chen et al., 2016).

*been conducted on the correlation between the South Asian climate/weather systems and the Eurasian snow".*

⇒ These two sentences are now merged in the revised manuscript as the following:

*"Furthermore, the Eurasian snow is highly related with the climate and weather systems in Asia, as demonstrated by many studies."*

6. *P. 2, L. 11: Explain the interaction mechanism mentioned here briefly.*

⇒ Generally, extensive Eurasian snow cover during winter and spring tends to lead less rainfall during the Indian summer monsoon. An extensive snow cover over Eurasia during winter reduces lower tropospheric temperature. Then the cooling center which corresponds to the center of cyclone appears northern part of Eurasia, leading to a Rossby-wave-train-like circulation. This pattern propagates atmospheric disturbances induced by snow cover anomaly from Eurasia to Asia. To conclude, northerly wind over the East Asia leads to weaker summer monsoon rainfall (Liu and Yanai, 2002).

7. *P. 2, L. 14 ∼ 16: What is the key findings of these studies on the "interactions"?*

⇒ Generally, there are inverse relationship between the East Asian climate/weather systems and the Eurasian snow. However, depending on the target regions the relationship patterns tend to be different as the followings:

(a) **Yang and Xu (1994)** studied relationship between the Eurasian winter snow and summer rainfall over China. They concluded that interactions appear differently depending on the target regions. When they averaged all the rainfall over China as a whole, only a weak relationship appeared. However, when they divided the regions, the southern and northern China represent a strong in-phase relationship with the Eurasia winter snow while the western, central, and north-eastern parts of China showed reverse relationship with the Eurasia winter snow.

(b) **Kripalani et al. (2002)** studied the interactions between the winter-spring time Soviet snow and the Korean rainfall. They divided Eurasia in two section, one for positively correlated area and the other for negatively correlated area for both the Indian and Korean monsoon rainfalls. They found the most significant area: the winter/spring time snow depth over western Eurasia (over Kazakhstan) is negatively related, whereas the snow depth over eastern Eurasia (over eastern Siberia) is positively related with Korean monsoon rainfall.

(c) **Wu and Qian (2003)** studied the relationship between the Tibetan Plateau (TP) winter snow and the Asian summer monsoon by using the observations over TP. They performed empirical orthogonal function (EOF) analysis and defined 3 different patterns: 1) light snow over the entire TP; 2) heavy snow over the eastern TP; and 3) heavy snow over the southwestern TP. For the second and the third mode, the south and southeast Asian summer monsoon becomes weak leading to less summer rainfall, whereas the first mode represents opposite phase.

(d) **Zhao et al. (2007)** studied the relationship between the spring snow over TP and hemispheric extratropical circulation/East Asian summer monsoon rainfall. They concluded that the increase of the spring snow is associated with decreases of local tropospheric temperature and geopotential height in the spring and the early summer. Also, increase of the spring snow significantly decreases 500-mb geopotential height and makes anomalous northeasterlies which weakens the East Asian summer monsoon, leading to a decrease of surface air temperature and rainfall at the Yangtze and Hwai Rivers and an increase of rainfall in the southeastern China.

(e) **Won et al. (2017)** studied about the relationship between the Eurasian snow cover pattern and the variation of the Korean summer temperature. They also performed the EOF analysis, and found that the first EOF mode represented the zonally elongated pattern and second EOF mode represented the east-west dipole-like pattern of snow cover. They found that the first mode, which is a zonally elongated pattern of snow cover over the whole Eurasian region, is more correlated with the Korean temperature during June while the dipole pattern is related with the Korean temperature during August.

8. *P. 2, L. 30: The 1st goal sounds strange. The developers as well as most users of these LSMs might already understand them.*

⇒ We also think that the developers definitely understand the LSMs. However, even though our study does not provide complete and detailed information on all the existing LSMs, we still believe that our 1st goal can provide useful information at least to many users and/or the readers who are not familiar with these models.

9. *Section 2.1: Consider to add a table describing key differences in snow physical processes incorporated in the LSMs.*

⇒ We added a table describing key differences in the LSMs as in Table R1 below:

**Table R1.** The general feature of each LSM.

| Snow | Noah LSM (Liveh et al., 2010) | Noah MP (Niu et al., 2010) | RUC LSM (Benjamin et al., 2004) | CLM4 (Oleson et al., 2010) |
|---|---|---|---|---|
| Layers | 1 | 3 | 2 | 5 |
| Density | Fixed | Calculates variable snow density | Calculates variable snow density | Calculates variable snow density |
| Liquid | X | O | O | O |

10. *P.4, L. 1 ∼ 2: Indicate directions of energy fluxes mentioned here.*

⇒ In the revised manuscript, we have described the directions of energy fluxes by modifying equation (2) in the original manuscript and provided supplementary explanations on the meaning of each element as the following:

$$M_s = \frac{1}{L}\left(Q_{sw\downarrow} - Q_{sw\uparrow} + Q_{lw\downarrow} - Q_{lw\uparrow} - Q_{lwo} - Q_{lt} - Q_{sn} - Q_g\right) \tag{1}$$

where $M_s$ is snowmelt rate, $Q_{sw\downarrow}$ is incoming solar radiation, $Q_{sw\uparrow}$ is reflected solar radiation, $Q_{lw\downarrow}$ is incoming longwave radiation, $Q_{lw\uparrow}$ is reflected longwave radiation, $Q_{lwo}$ is outgoing longwave radiation, $Q_{lt}$ is latent heat flux due to vaporization of water, $Q_{sn}$ is sensible heat flux, and $Q_g$ is heat flux into the soil.

11. *P. 4, L. 11: "tuning parameter" of what?*

   ⇒ We meant the tuning parameter for the snow melting rate. We explained about this tuning parameter in detail in #13 below.

12. *P. 4, L. 30: What do the authors mean by "destructive metamorphism"?*

   ⇒ We tried to indicate the phenomena of snow structure transformation by the word "destructive metamorphism". To mention briefly, the transformation of snow structure can vary depending on the ambient conditions. For example, under the variable temperature gradients, snow keeps recrystallizing by repeating melting and refreezing. The recrystallizing causes the snow crystals to be larger and less coherent (see more details in `https://www.slf.ch/en/snow/snow-as-a-material/snow-metamorphism.html`). Therefore, the Noah-MP can consider destructive metamorphism since it has multiple snow layer which can simulate the temperature gradient through the layers.

13. *P. 4, L. 33: "Melting factor" might be a tuning parameter. What effects can we expect from the modulation of this parameter?*

   ⇒ The Noah-MP calculates fractional snow cover (SCF) as (Niu and Yang, 2007):

$$
\begin{aligned}
SCF &= \tanh\left[\frac{h_{sno}}{2.5 Z_{0g} f_{melt}}\right] \\
f_{melt} &= \left(\frac{\rho_{sno}}{\rho_{new}}\right)^m
\end{aligned}
\tag{2}
$$

where $h_{sno}$ is the snow depth, $Z_{0g}$ is the ground roughness, $f_{melt}$ is melting factor for snow cover fraction, $\rho_{new}$ is fresh snow density which is used as 100 kg/m$^3$ to scale the actual snow depth, $\rho_{sno}$. $m$ is a melting factor determining the curves in the melting season. The melting factor is calibrated based on observations and according to the technical note of the Noah-MP, and it is generally larger for larger scale. In the WRFV3.6.1, which is used in this study, the Noah-MP fixed melting factor to 2.5 for every land-use type. However, as the snow melting rate varies depending on the region, modulating the melting factor based on the observation over the target region will contribute to more accurate prediction. For example, Tomasi et al. (2017) modulated the melting factor to 1.0 which resulted in reasonable SCF compared to satellite observation over the reference station.

14. *P. 5, L. 6 ∼ 8: The explanation on "snow age" is difficult to understand. Why do the authors relate snow age to grain growth effect and soot effect?*

⇒ We related snow age to the grain growth effect and the soot effect because the Noah-MP calculates snow age by considering the effects of grain growth due to vapor diffusion, the effects of grain growth at freezing of melt water, and the effects of soot as the following:

$$
\begin{aligned}
A_1 &= e^{5 \times 10^3 \left( \frac{1}{T_{frz}} - \frac{1}{T_g} \right)} \\
A_2 &= e^{5 \times 10^4 \left( \frac{1}{T_{frz}} - \frac{1}{T_g} \right)} \\
A_3 &= 0.3 \\
A_t &= A_1 + A_2 + A_3
\end{aligned}
\tag{3}
$$

where $A_1$ represents the effect of grain growth due to vapor diffusion, $A_2$ represents the effects of grain growth at freezing of melt water, $A_3$ represents the soot effect, and $A_t$ represents the total effect of these three. Then snow age, $S_a$ is calculated by using $A_t$ (more details in Noah-MP technical description note):

$$
\begin{aligned}
S_a &= \frac{\tau_{ss}}{\tau_{ss} + 1} \\
\tau_{ss} &= \left( \tau_{ss} + 1 \times 10^{-6} dt A_t \right) \left( 1 - S_w + S_m \right).
\end{aligned}
\tag{4}
$$

15. *P. 6, L. 3: Change 2.5 K to 275.65 K*

⇒ We appreciate the referee checking this out. We modified it accordingly in the revised manuscript.

16. *P.6, L. 15: How often did the authors perform initialization of the atmosphere and land? Only at the beginning of the calculation period (1 June 2009) (climate simulation mode) or every day (weather forecast mode)?*

⇒ We initialized the atmosphere and land only at the beginning while the boundary conditions were updated every 6 hours through the FNL data. For the land, we conducted initialization only for the surface layer excluding the subsurface variables.

17. *P. 6, L. 25: It is not necessary to mention vorticity and ozone here.*

⇒ We agree with the referee in the viewpoint that we didn't use tropospheric ozone and vorticity to initialize the model. However, we want to keep the original information on the FNL data since P. 6, L. 25 is a part of data description. Instead, we modified this part by clarifying the variables actually used for initialization as the following:

"In our study, we actually used temperature, horizontal wind component, relative humidity, geopotential height, surface pressure, sea-level pressure, ice flag, and soil variables to initialize the model."

18. *P. 7, L. 3: What is "snow-climate classes"? Please explain more.*

⇒ Sturm et al. (1995) classified the snow into 6 classes: tundra, taiga, Alpine, maritime, ephemeral, and prairie. To calculate SWE, CMC followed this classification and averaged by monthly from October to June assuming that the Canadian density observations are representative of snow-climate classes in other regions of the Northern Hemisphere. Here, the monthly values for the 6 different snow classes (Table R2 below) are called the "snow-climate classes" (more details in https://nsidc.org/data/nsidc-0447):

**Table R2.** Monthly mean snow density over the snow-climate classes defined by Sturm et al. (1995).

| Month | Tundra | Taiga | Alpine | Maritime | Ephemeral | Prairie |
|-------|--------|-------|--------|----------|-----------|---------|
| Oct | 200.0 | 160.0 | 160.0 | 250.0 | 140.0 | 160.0 |
| Nov | 210.7 | 176.9 | 183.5 | 300.0 | 161.6 | 172.0 |
| Dec | 218.1 | 179.8 | 197.7 | 335.1 | 185.1 | 181.6 |
| Jan | 230.3 | 193.1 | 216.5 | 316.8 | 213.7 | 207.2 |
| Feb | 242.7 | 205.9 | 248.5 | 337.3 | 241.6 | 241.5 |
| Mar | 254.4 | 221.8 | 283.3 | 364.3 | 261.0 | 263.5 |
| Apr | 273.6 | 263.2 | 332.0 | 404.6 | 308.0 | 312.0 |
| May | 311.7 | 319.0 | 396.3 | 458.6 | 398.1 | 399.6 |
| Jun | 369.3 | 393.4 | 501.0 | 509.8 | 464.5 | 488.9 |

19. *P. 7, L. 11: What does the "differences" tell us? Please describe.*

⇒ The NDSI is a useful method to measure the reflectance differences between visible (green) and shortwave infrared by controlling variance of two bands. This is useful for snow since snow tends to reflect visible part of the electromagnetic spectrum and absorb the NIR or the short-wave infrared part of the spectrum, while cloud mostly show high reflectivity. Therefore, this method allows to separate clouds and snow effectively (more details in `https://eos.com/ndsi/`).

20. *P. 7, L. 30: The intention of "microphysical options" is not clear. Please detail more.*

⇒ The Noah-MP has multi-microphysics options as follows (detailed equations in the Noah-MP technical description note):

   (a) **Leaf area index**
       - Use climatology leaf area index (prescribed)
       - Calculate leaf area index (predicted)
   (b) **Turbulent transfer**
       - Original Noah
       - NCAR LSM
   (c) **Soil moisture stress factor for transpiration**
       - Noah type using soil moisture
       - CLM type using matric potential
       - SSiB type also using matric potential but expressed by a different function
   (d) **Canopy stomal resistance**
       - Bell-Berry stomatal conductance scheme
       - Jarvis stomatal resistance scheme
   (e) **Snow surface albedo**
       - BATS
       - CLASS
   (f) **Frozen soil permeability**

- – Linear effect, more permeable scheme from Niu and Yang (2006)
- – Nonlinear effects, less permeable scheme from original Noah

(g) **Supercooled liquid water**
- – No iteration from Niu and Yang (2006)
- – Koren's iteration (Koren et al., 2006) with Flerchingers explicit solution

(h) **Radiation transfer**
- – Modified two-stream radiation-transfer scheme
- – Two-stream applied to the entire cell
- – Two-stream applied to fractional vegetated area

(i) **Partitioning of precipitation to snowfall and rainfall**
- – Based on Jordan (1991)

$$F_{p,ice} = \begin{cases} 0 & T_{sfc} > T_{frz} + 2.5 \\ 1 & T_{sfc} \leq T_{frz} + 0.5 \\ 1 - (-54.632 + 0.2 \times T_{sfc}) & T_{frz} + 0.5 < T_{sfc} \leq T_{frz} + 2 \\ 0.6 & T_{frz} + 2 < T_{sfc} \leq T_{frz} + 2.5 \end{cases} \tag{5}$$

- – Based on BATS when $T_{sfc} = T_{frz} + 2.5$

$$F_{p,ice} = \begin{cases} 0 & T_{sfc} \geq T_{frz} + 2.2 \\ 1 & T_{sfc} < T_{frz} \end{cases} \tag{6}$$

- – Based on when $T_{sfc} = T_{frz}$

$$F_{p,ice} = \begin{cases} 0 & T_{sfc} \geq T_{frz} \\ 1 & T_{sfc} < T_{frz} \end{cases} \tag{7}$$

(j) **Runoff and ground water**
- – TOPMODEL with groundwater (Niu et al., 2007)
- – TOPMODEL with an equilibrium water table (Niu et al., 2005)
- – Original surface and subsurface runoff (free drainage)
- – BATS surface and subsurface runoff (free drainage)

**References**

Abdallah, A. A., M. M. Eid, M. M. A. Wahab, and F. M. El-Hussainy: Regional climate simulation of WRF model over North Africa: Temperature and precipitation, World Environ., 5, 160–173, 2015.

Abramowitz, G. R. Leuning, M. Clark, and A. Pitman: Evaluating the performance of land surface models, J. Clim., 21, 5468-5481, `https://doi.org/10.1175/2008JCLI2378.1`, 2008.

Allen, R. J., and C. S. Zender: Forcing of the Arctic Oscillation by Eurasian snow cover, J. Clim., 24, 6528-6539, `https://doi.org/10.1175/2011JCLI4157.1`, 2011.

Alsarraf, H., and M. van den Broeke: Using the WRF regional climate model to simulate future summertime wind speed changes over the Arabian Peninsula, J. Climatol. Wea. Forecasting, 3, 144, `doi:10.4172/2332-2594.1000144`, 2015.

Bamzai, A. S., and J. Shukla: Relation between Eurasian snow cover, snow depth, and the Indian summer monsoon: An observational study, J. clim., 12, 3117-3132, `https://doi.org/10.1175/1520-0442(1999)012<3117:RBESCS>2.0.CO;2`, 1999.

Bernett, T. P., L. Dmenil, U. Schlese, E. Roeckner, and M. Latif: The effect of Eurasian snow cover on regional and global climate variations, J. Atmos. Sci., 46, 661-686, `https://doi.org/10.1175/1520-0469(1989)046<0661:TEOESC>2.0.CO;2`, 1989.

Bukovsky, M. S., and D. J. Karoly: Precipitation simulations using WRF as a nested regional climate model, J. Appl. Meteorol. Climatol., 48, 2152–2159, 2009.

Chen, X., S. Liang, and Y. Cao: Satellite observed changes in the Northern Hemisphere snow cover phenology and the associated radiative forcing and feedback between 1982 and 2013, Environ. Res. Lett., 11, 084002, `https://doi.org/10.1088/1748-9326/11/8/084002`, 2016.

Chervenkov, H., T. Todorov, and K. Slavov: Snow cover assessment with regional climate model – Problems and results, in: Large-scale Scientific Computing, Lirkov, I., S. D. Margenov, and J. Wasniewski (Eds.), LSSC 2015, LNCS 9374, 327-334, `doi:10.1007/978-3-319-26520-9_36`, 2015.

Choi, M., S. O. Lee, and H. Kwon: Understanding of the common land model performance for water and energy fluxes in a farmland during the growing season in Korea, Hydrol. Process., 24, 1063-1071, `https://doi.org/10.1002/hyp.7567`, 2010.

Chotamonsak. C., E. P. Salathé, Jr., J. Kreasuwan, S. Chantara, and K. Siriwitayakorn: Projected climate change over Southeast Asia simulated using a WRF regional climate model, Atmos. Sci. Lett., 12, 213–219, 2011.

Chou, C.: Landsea heating contrast in an idealized Asian summer monsoon, Clim. Dynam., 21, 11-25, `https://doi.org/10.1007/s00382-003-0315-7`, 2003.

Cohen, J., and D. Entekhabi: Eurasian snow cover variability and Northern Hemisphere climate predictability, Geophys. Res. Lett., 26, 345-348, `https://doi.org/10.1029/1998GL900321`, 1999.

Cooper, M. J., R. V. Martin, A. I. Lyapustin, and C. A. McLinden: Assessing snow extent data sets over North America to inform and improve trace gas retrievals from solar backscatter, Atmospheric Meas. Tech., 11, 2983-2994, `https://doi.org/10.5194/amt-11-2983-2018`, 2018.

Dasari, H. P., R. Salgado, J. Perdigao, and V. S. Challa: A regional climate simulation study using WRF-ARW model over Europe and evaluation for extreme temperature weather events, Int. J. Atmos. Sci., 2014, 704079, `http://dx.doi.org/10.1155/2014/704079`, 2014.

Dawson, N., P. Broxton, X. Zeng, M. Leuthold, M. Barlage, and P. Holbrook: An evaluation of snow initializations in NCEP global and regional forecasting models, J. Hydrometeor., 17, 1885-1901, 2016.

Drusch, M., D. Vasiljevic, and P. Viterbo: ECMWFs global snow analysis: Assessment and revision based on satellite observations, J. Appl. Meteorol. Climatol., 43, 1282-1294, `https://doi.org/10.1175/1520-0450(2004)043<1282:EGSAAA>2.0.CO;2`, 2004.

Etchevers, P., and coauthors: Validation of the energy budget of an alpine snowpack simulated by several snow models (SnowMIP project), Ann. Glaciol., 38, 150-158, `doi:10.3189/172756404781814825`, 2004.

Gilliam, R. C., and J. E. Pleim: Performance assessment of new land surface and planetary boundary layer physics in the WRF-ARW, J. Appl. Meteorol. Climatol., 49, 760-774, `https://doi.org/10.1175/2009JAMC2126.1`, 2010.

Hall, D. K., and G. A. Riggs: Accuracy assessment of the MODIS snow products, Hydrol. Process., 21, 1534-1547, `https://doi.org/10.1002/hyp.6715`, 2007.

Hall, D. K., G. A. Riggs, J. L. Foster, and S. V. Kumar: Development and evaluation of a cloud-gap-filled MODIS daily snow-cover product, Remote Sens. Environ., 114, 496-503, 2010.

Hancock, S., R. Baxter, J. Evans, and B. Huntley: Evaluating global snow water equivalent products for testing land surface models, Remote Sens. Environ., 128, 107-117, `https://doi.org/10.1016/j.rse.2012.10.004`, 2013.

Harding, K. J., P. K. Snyder, and S. Liess: Use of dynamical downscaling to improve the simulation of CentralU.S. warm season precipitation in CMIP5 models, J. Geophys. Res. Atmos., 118, 12522-12536, `doi:10.1002/2013JD019994`, 2013.

Henderson-Sellers, A., Z.-L. Yang, and R. E. Dickinson: The Project for Intercomparison of Land-Surface Parameterization Schemes (PILPS), Bull. Amer. Meteor. Soc., 74, 1335-1349, 1993.

Henderson-Sellers, A., A. J. Pitman, P. K. Love, P. Irannejad, and T. H. Chen: The Project for Intercomparison of Land-Surface Parameterization Schemes (PILPS): Phases 2 and 3, Bull. Amer. Meteor. Soc., 76, 489-503, 1995.

Henderson-Sellers, A, K. McGuffie, and A. J. Pitman: The Project for Intercomparison of Land-surface Parameterization Schemes (PILPS): 1992 to 1995. Clim. Dyn., 12, 849-859, 1996.

Jacob, D., and coauthors: An inter-comparison of regional climate models for Europe: model performance in present-day climate, Clim. Change, 81, 31-52, `https://doi.org/10.1007/s10584-006-9213-4`, 2007.

Jin, J., N. L. Miller, and N. Schlegel: Sensitivity study of four land surface schemes in the WRF model, Adv. Meteorol. 2010, `https://doi.org/10.1155/2010/167436`, 2010.

Kamae, Y., M. Watanabe, M. Kimoto, and H. Shiogama: Summertime landsea thermal contrast and atmospheric circulation over East Asia in a warming climatePart I: Past changes and future projections, Cilm. Dynam., 43, 2553-2568, `https://doi.org/10.1007/s00382-014-2073-0`, 2014.

Koo, M. S., S. Baek, K. H. Seol, and K. Cho: Advances in land modeling of KIAPS based on the Noah Land Surface Model, Asia-Pac. J. Atmospheric Sci., 53, 361-373, `https://doi.org/10.1007/s13143-017-0043-2`, 2017.

Krinner, G., and coauthors: ESM-SnowMIP: assessing snow models and quantifying snow-related climate feedbacks, Geosci. Model Dev., 11, 5027-5049, doi:10.5194/gmd-11-5027- 2018, 2018.

Kripalani, R. H., B. J. Kim, J. H. Oh., and S. E. Moon: Relationship between Soviet snow and Korean rainfall, Int. J. Climatol., 22, 1313-1325, `https://doi.org/10.1002/joc.809`, 2002.

Kumar, S., and Coauthors: Assimilation of remotely sensed soil moisture and snow depth retrievals for drought estimation, J. Hydrometeor., 15, 2446-2469, `https://doi.org/10.1175/JHM-D-13-0132.1`, 2014.

Kumar, S., C. D. Peters-Lidar, K. R. Arsenault, A. Getirana, D. Mocko, and Y. Liu: Quantifying the added value of snow cover area observations in passive microwave snow depth data assimilation, J. Hydrometeor, 16, 1736-1741, `https://doi.org/10.1175/JHM-D-15-0021.1`, 2015.

Li, C. and M. Yanai: The onset and interannual variability of the Asian summer monsoon in relation to landsea thermal contrast, J. Clim., 9, 358-375, `https://doi.org/10.1175/1520-0442(1996)009<0358:TOAIVO>2.0.CO;2`, 1996.

Liang, X., E. F. Wood, and D. P. Lettenmaier: Modeling ground heat flux in land surface parameterization schemes, J. Geophys. Res., 104, 9581-9600, `https://doi.org/10.1029/98JD02307`, 1999.

Liu, J., and coauthors: Validation of Moderate Resolution Imaging Spectroradiometer (MODIS) albedo retrieval algorithm: Dependence of albedo on solar zenith angle, J. Geophys. Res. Atmos., 114, D01106, `https://doi.org/10.1029/2008JD009969`, 2009.

Liu, X., and M. Yanai: Influence of Eurasian spring snow cover on Asian summer rainfall, Int. J. Climatol., 22, 1075-1089, `https://doi.org/10.1002/joc.784`, 2002.

Liu, Y., C. D. Peters-Lidard, S. Kumar, J. L. Foster, M. Shaw, Y. Tian, and G. M. Fall: Assimilating satellite-based snow depth and snow cover products for improving snow predictions in Alaska, Adv. Water Resour., 54, 208-227, `https://doi.org/10.1016/j.advwatres.2013.02.005`, 2013.

Moriasi, D. N., J. G. Arnold, M. W. Van Liew, R. L. Bingner, R. D. Harmel, and T. L. Veith: Model evaluation guidelines for systematic quantification of accuracy in watershed simulations, Trans. ASABE, 50, 885-900, `https://doi.org/`, 2007.

Mooney, P. A., F. J. Mulligan, C. L. Bruyère, C. L. Parker, D. O. Gill: Investigating the performance of coupled WRF-ROMS simulations of Hurricane Irene (2011) in a regional climate modeling framework, Atmos. Res., 215, 57–74, 2019.

Nester, T., R. Kirnbauer, J. Parajka, and G. Bloschl: Evaluating the snow component of a flood forecasting model, Hydrol. Res., 43, 762-779, `https://doi.org/10.2166/nh.2012.041`, 2012.

Niu, G. Y., and Z. L. Yang: An observation-based formulation of snow cover fraction and its evaluation over large North American river basins, J. Geophys. Res. Atmos., 112, D21101, `https://doi.org/10.1029/2007JD008674`, 2007.

Oaida, C. M., Y. Xue, M. G. Flanner, S. M. Skiles, F. de Sales, and T. H. Painter: Improving snow albedo processes in WRF/SSiB regional climate model to assess impact of dust and black carbon in snow on surface energy balance and hydrology over western U.S., J. Geophys. Res. Atmos., 120, 3228–3248, 2015.

Parajka, J., and G. Bloschl: The value of MODIS snow cover data in validating and calibrating conceptual hydrologic models, J. Hydrol., 358, 240-258, 2008.

Pitman, A. J., and A. Henderson-Sellers: Recent progress and results from the project for the intercomparison of land surface parameterization schemes. J. Hydrol., 212-213, 128-135, 1998.

Qi, L., J. He, Z. Zhang, and J. Song: Seasonal cycle of the zonal land-sea thermal contrast and East Asian subtropical monsoon circulation, Chin. Sci. Bull., 53, 131-136, 2008.

Raghavan, S. V., M. T. Vu, and S. Y. Liong: Regional climate simulations over Vietnam using the WRF model, Theor. Appl. Climatol., 126, 161–182, 2016.

Ramarohetra, J., B. Pohl, and B. Sultan: Errors and uncertainties introduced by a regional climate model in climate impact assessments: example of crop yield simulations in West Africa, Environ. Res. Lett., 10, 124014, `doi:10.1088/1748-9326/10/12/124014`, 2015.

Ratna, S. B., J. V. Ratnam, S. K. Behera, F. T. Tangang, and T. Yamagata: Validation of the WRF regional climate model over the subregions of Southeast Asia: climatology and interannual variability, Clim. Res., 71, 263–280, 2017.

Reichle, R. H., R. D. Koster, G. J. M. Lannoy, B. A. Forman, Q. Liu, and S. P. P. Mahanama: Assessment and enhancement of MERRA land surface hydrology estimates, J. Climate, 24, 6322-6338, `https://doi.org/10.1175/JCLI-D-10-050331.1`, 2011.

Riggs, G. A., D. K. Hall, and M. O. Roman: Overview of NASAs MODIS and visible infrared imaging radiometer suite (VIIRS) snow-cover earth system data records, Earth Syst. Sci. Data, 9, 765-777, `https://doi.org/10.5194/essd-9-765-2017`, 2017.

Rodell, M., and P. R. Houser: Updating a land surface model with MODIS-derived snow cover, J. Hydrometeor., 5, 1064-1075, `https://doi.org/10.1175/JHM-395.1`, 2004.

Roxy, M. K., K. Ritika, P. Terray, R. Murtugudde, K. Ashok, and B. N. Goswami: Drying of Indian subcontinent by rapid Indian Ocean warming and a weakening land-sea thermal gradient, Nat. Commun., 6, 7423, `https://doi.org/10.1038/ncomms8432`, 2015.

Sheffield, J., and coauthors: North American climate in CMIP5 experiments. Part I: Evaluation of historical simulations of continental and regional climatology, J. Clim., 26, 9209-9245, `doi:10.1175/JCLI-D-12-00592.1`, 2013.

Sippel, S., J. Zscheischler, M. D. Mahecha, R. Orth, M. Reichstein, M. Vogel, and S. I. Seneviratne: Refining multi-model projections of temperature extremes by evaluation against land-atmosphere coupling diagnostics, Earth Syst. Dynam., 8, 387-403, `doi:10.5194/esd-8-387-2017`, 2017.

Steiner, A. L., J. Pal, F. Giorgi, R. E. Dickinson, W. L. Chameides: Coupling of the Common Land Model (CLM0) to a regional climate model (RegCM). Theor. Appl. Climatol. 82, 225-243, `doi:10.1007/s00704-005-0132-5`, 2005.

Sun, Y., Y. Ding, and A. Dai: Changing links between South Asian summer monsoon circulation and tropospheric landsea thermal contrasts under a warming scenario, Geophys. Res. Lett. 37, L02704, `https://doi.org/1029/2009GL041662`, 2010.

Tao, M., L. Chen, Z. Wang, J. Tao, H. Che, X. Wang, and Y. Wang: Comparison and evaluation of the MODIS Collection 6 aerosol data in China, J. Geophys. Res. Atmos., 120, 6992-7005, `https://doi.org/10.1002/2015JD023360`, 2015.

Tomasi, E., L. Giovannini, D. Zardi, and M. de Franceschi: Optimization of noah and noah-MP WRF land surface schemes in snow-melting conditions over complex terrain, Mon. Wea. Rev., 145, 4727-4745, `https://doi.org/10.1175/MWR-D-16-0408.1`, 2017.

Toure, A. M., Rodell, M., Yang, Z. L., Beaudoing, H., Kim, E., Zhang, Y., and Kwon, Y.: Evaluation of the snow simulations from the community land model, version 4 (CLM4), J. Hydrometeorol., 17, 153-170, `https://doi.org/10.1175/JHM-D-14-0165.1`,2016.

Turrent, C. and T. Cavazos: Role of the land-sea thermal contrast in the interannual modulation of the North American Monsoon, Geophys. Res. Lett., 36, L02808, `https://doi.org/10.1029/2008GL036299`, 2009.

Ukkola, A. M., M. G. De Kauwe, A. J. Pitman, M. J. Best, G. Abramowitz, V. Haverd, M. Decker, and N. Haughton: Land surface models systematically overestimate the intensity, duration and magnitude of seasonal-scale evaporative droughts, Environ. Res. Lett., 11, 104012, `doi:10.1088/1748-9326/11/10/104012`, 2016.

Walsh, J. E., W. L. Chapman, V. Romanovsky, J. H. Christensen, M. Stendel: Global climate model performance over Alaska and Greenland, J. Clim., 21, 6156-6174, `https://doi.org/10.1175/2008JCLI2163.1`, 2008.

Wu, B., K. Yang, and R. Zhang: Eurasian snow cover variability and its association with summer rainfall in China, Adv. Atmospheric Sci., 26, 31-44, `https://doi.org/10.1007/s00376-009-0031-2`, 2009.

Wu. G, Y. Liu, B. He, Q. Bao, A. Duan, and F. F. Jin: Thermal controls on the Asian summer monsoon, Sci. Rep., 2, 404, `https://doi.org/10.1038/srep00404`, 2012.

Wu, R., and B. P. Kirtman: Observed relationship of spring and summer East Asian rainfall with winter and spring Eurasian snow, J. Clim., 20, 1285-1304, `https://doi.org/10.1175/JCL14068.1`, 2007.

Yang, Z.-L., and Coauthors: The community Noah land surface model with multiparameterization options (Noah-MP): 2. Evaluation over global river basins. J. Geophys. Res., 116, D12100, `https://doi.org/10.1029/2010JD015140`, 2011.

Yuan, S., and S. M. Quiring: Evaluation of soil moisture in CMIP5 simulations over the contiguous United States using in situ and satellite observations, Hydrol. Earth Syst. Sci., 21, 2203-2218, `doi:10.5194/hess-21-2203-2017`, 2017.

Zhang, C., Y. Wang, A. Lauer, and K. Hamilton: Configuration and evaluation of the WRF model for the study of Hawaiian regional climate, Mon. Wea. Rev., 140, 3259–3277, 2012.

Zhou, X., and Coauthors: Evaluation of Arctic land snow cover characteristics, surface albedo, and temperature during the transition seasons from regional climate model simulations and satellite data, Adv. Meteorol., `https://doi.org/10.1155/2014/604157`, 2014.

---

## Author Comment (AC2) · 8 Apr 2019

**Reply to the Comments by Referee #2 for Manuscript tc-2019-15**

*In this study, the authors tried to address the uncertainty in predicting the Eurasian snow by inter-comparing the performances of four land surface models coupled to WRF. They have four goals to achieve but the purpose of the work is not clearly articulated. The manuscript can be accepted to be published only after some major concerns have been addressed:*

⇒ The authors appreciate the referee's valuable comments on this study, which made significant improvement in the manuscript. We have made our best effort to revise the manuscript based on the referee's comments and suggestions. In the following, we made an item-by-item response to the specific comments by the referee.

(1) *Many studies show errors in the input and validation data, rather than model formulation, seem to be the greatest factor affecting model performance. The manuscript lacks of discussion of the quality for those "observation" used to evaluate the model performance.*

⇒ We appreciate the referee pointing this out. We used these data because they have been widely used for the validation purpose as seen in many studies. Thus we assume that we can use these data without further quality check. We added a description on the quality of observations used in our study, based on the followings:

1) Cooper et al. (2018) assessed snow extent data sets including both CMC data and MODIS Terra data by assuming the surface observations are truth. They compared seven snow extent data sets – CMC, IMS, MAIAC AQUA, MAIAC TERRA, MODIS AQUA, MODIS TERRA, and NISE. They found that CMC represented strong agreement with in situ observations with $F$ score 0.81 while MODIS TERRA represented medium agreement with $F$ score 0.54. More specifically, they calculated accuracy, precision, recall, and $F$ score as the following:

$$
\begin{aligned}
\text{Accuracy} &= \frac{TP + TN}{TP + TN + FP + FN} \\
\text{Precision} &= \frac{TP}{TP + FP} \\
\text{Recall} &= \frac{TP}{TP + FN} \\
F &= 2 \times \frac{\text{precision} \times \text{recall}}{\text{precision} + \text{recall}}
\end{aligned}
\tag{1}
$$

where $TP$ is true positive, $TN$ is true negative, $FP$ is false positive, and $FN$ is false negative. The assessment results are summarized in Table R1.

2) The Canadian Meteorological Centre (CMC) snow depth data is the most widely used data in many model evaluation studies, being considered as the surface truth value (e.g., Niu and Yang 2007; Reichle et al., 2011; Yang et al., 2011; Liu et al., 2013; Kumar et al., 2014, 2015; Zhou et al., 2014; Dawson et al., 2016; Toure et al., 2016).

3) The MODIS/Terra Snow Cover Monthly L3 Global 0.05Deg CMG, Version 6 is also very widely used data in validating many model outputs and improving model

**Table R1.** Evaluation of daily snow extent data set performance for 2015. The GHCN-D surface observations are used as "truth". All products are regridded to a common 4 km resolution. The highest value for each metric is shown in bold. From Cooper et al. (2018).

|  | Accuracy | Precision | Recall | $F$ |
|---|---|---|---|---|
| CMC | 0.91 | 0.79 | 0.83 | 0.81 |
| IMS | 0.93 | 0.87 | 0.83 | 0.85 |
| MAIAC AQUA | 0.91 | 0.90 | 0.74 | 0.82 |
| MAIAC TERRA | 0.91 | 0.90 | 0.75 | 0.82 |
| MODIS AQUA | 0.76 | 0.51 | 0.43 | 0.46 |
| MODIS TERRA | 0.82 | 0.69 | 0.45 | 0.54 |
| NISE | 0.84 | 0.83 | 0.45 | 0.58 |

prediction skill (Drusch et al., 2004; Rodell and Houser, 2004; Parajka and Bloschl, 2008; Hall et al., 2010; Nester et al., 2012; Hancock et al., 2013; Zhou et al., 2014).

(2) *Among four goals of this paper only the third one has been addressed sufficiently. The first two haven't been explained clearly. The reader can easily understand the difference between four models if a table is used to show the different treatment of snow albedo, snow density, snow compaction, snow interception, snow age, and etc. The authors need to make some sensitivity tests like the different microphysics schemes to address and make any conclusion on the last goal of this paper.*

⇒ We appreciate these valuable suggestions by the referee. Following the reviewer's suggestion, we have added a table in the revised manuscript to show the characteristics of each model (see Table R2 below):

**Table R2.** The general feature of each LSM.

| Snow | Noah LSM (Liveh et al., 2010) | Noah MP (Niu et al., 2010) | RUC LSM (Benjamin et al., 2004) | CLM4 (Oleson et al., 2010) |
|---|---|---|---|---|
| Layers | 1 | 3 | 2 | 5 |
| Density | Fixed | Calculates variable snow density | Calculates variable snow density | Calculates variable snow density |
| Liquid | X | O | O | O |

⇒ We have also followed the referee's suggestion to do some sensitivity tests, e.g., using different microphysics schemes, to address the last goal of this paper. Although we could not perform extensive tests, we have done some sensitivity experiments using other microphysics scheme. For example, the present study employed the WSM3 scheme, and we conducted sensitivity experiments using the WSM6 scheme. The results indicated that all the LSMs represented better performances with WSM6 than WSM3. We have included such sensitivity results and discussions in the revised manuscript to address the last goal of this study.

(3) *For fair inter-comparison the forcing should be as close as possible. For snow predictions how to differenciate snowfall or rainfall from the total precipitation is critical. This process should be mainly determined by microphysics scheme. The authors should use the same MP for all models to determine the amount of snowfall and rainfall rather than the empirical method by each model.*

⇒ We agree with the referee on the statement "*For snow predictions, how to differenciate snowfall or rainfall from the total precipitation is critical*". But we cannot fully accept the referee's suggestion to use the same MP for all models to determine the amount of snowfall/rainfall. We believe that each model has its own characteristics, including the way to classify the total precipitation into liquid versus solid forms. Although these schemes might be based on empirical methods, we believe that they are devised to balance with other physical/dynamical processes in the model; thus, formulating their own characteristic ways to calculate each process in the model. The referee's speculation may be right in terms of fair intercomparison; however, by doing so, we are afraid that the model balance would be broken and/or the model characteristics might be lost. Although we respect the referee's suggestion, it is another important thing in this study to examine the characteristic behavior of each model, when coupled to a regional climate model; thus, we decided to use the MP as it is in each model.

(4) *For the most part, snow models are built on similar principles. The greatest differences are found in how each model parameterizes individual processes (e.g., surface albedo and snow compaction). Parameterization choices naturally span a wide range of complexities. Ensemble is a promising way to reduce these uncertainties. It would be greatly beneficial to the title of this paper if the authors can also compare the performance of the ensemble mean of four models.*

⇒ We agree to the referee's comment. Due to the uncertainty involved in parameterizations, it is desirable to employ an ensemble approach. On the other hand, we think that the characteristics of each LSM, coupled to a regional climate model, may be better understood through deterministic approach and various sensitivity experiments. The referee's suggestion on the ensemble approach is fantastic, but it is left for a further study as the next step. We do appreciate this valuable advice by the referee.

(5) *Noah MP itself has so many options to choose. Many of them can have significant impact on the snow prediction. The authors should have optimal options before comparing it to the other three models.*

⇒ We think the referee pointed out an important issue. As the referee mentioned, the Noah-MP has many parameterization schemes in calculating leaf are index (dveg), surface layer drag coefficient (opt_sfc), canopy stomal resistance (opt_crs), snow surface albedo (opt_alb), frozen soil permeability (opt_inf), supercooled liquid water (opt_frz), radiation transfer (opt_rad), partitioning of precipitation to snowfall and rainfall (opt_snf) and, runoff and ground water (opt_run) (more details in the Noah-MP technical description note, `http://www.jsg.utexas.edu/noah-mp/files/Noah-MP_Technote_v0.2.pdf`).

We have performed some preliminary tests with multiple scheme combinations – 3 options in dveg, 2 options in opt_rad, and 1 option in opt_alb. Table R3 summarizes the best results from each scheme in terms of snow depth and RMSE. We noticed that the accuracy of snow depth prediction is highly influenced by the vegetation option compared to other options. Turning off the dynamic vegetation option and calculating the vegetation fraction inside Noah-MP results in the highest accuracy in predicting the Eurasian snow.

Note that finding the optimal scheme set requires an enormous amount of computing resources. The Noah-MP has 9 parameterization schemes, each having 2 4 options – 4 schemes with 2 options, 3 schemes with 3 options and 2 schemes with 4 options –

a total of 6912 combinations. This implies that we should run the model 6912 times to find the optimal set of parameterization schemes/options. In our study, as the Noah-MP is coupled to WRF, the computation time will tremendously increase to find the optimal scheme set, which is almost impossible. Therefore, we decided to adopt the default set of parameterization schemes/options in Noah-MP by assuming that this default set is the most general setting.

**Table R3.** The result of physical snow depth (m) and the RMSE of snow depth for each option. The name of options in this table "albedo", "radiation", "vegetation" mean that we modulated these options.

|  | Default | Albedo | Radiation | Vegetation |
|---|---|---|---|---|
| Snow Depth [m] | 0.200 (±0.060) | 0.203 (±0.062) | 0.193 (±0.058) | 0.193 (±0.058) |
| RMSE | 0.173 | 0.174 | 0.166 | 0.165 |

(6) *The quality of figure need to be improved. Some of them are very blurry and hard to read. Like figure 4a only 6 curves can be seen.*

⇒ We appreciate the careful check and suggestion by the referee on the quality of figures. We carefully checked all the figures and improved them by uploading the high-resolution files.

**References**

Cooper, M. J., R. V. Martin, A. I. Lyapustin, and C. A. McLinden: Assessing snow extent data sets over North America to inform and improve trace gas retrievals from solar backscatter, Atmospheric Meas. Tech., 11, 2983-2994, `https://doi.org/10.5194/amt-11-2983-2018`, 2018.

Dawson, N., P. Broxton, X. Zeng, M. Leuthold, M. Barlage, and P. Holbrook: An evaluation of snow initializations in NCEP global and regional forecasting models, J. Hydrometeor., 17, 1885-1901, 2016.

Drusch, M., D. Vasiljevic, and P. Viterbo: ECMWFs global snow analysis: Assessment and revision based on satellite observations, J. Appl. Meteorol. Climatol., 43, 1282-1294, `https://doi.org/10.1175/1520-0450(2004)043<1282:EGSAAA>2.0.CO;2`, 2004.

Hall, D. K., and G. A. Riggs: Accuracy assessment of the MODIS snow products, Hydrol. Process., 21, 1534-1547, `https://doi.org/10.1002/hyp.6715`, 2007.

Hall, D. K., G. A. Riggs, J. L. Foster, and S. V. Kumar: Development and evaluation of a cloud-gap-filled MODIS daily snow-cover product, Remote Sens. Environ., 114, 496-503, 2010.

Hancock, S., R. Baxter, J. Evans, and B. Huntley: Evaluating global snow water equivalent products for testing land surface models, Remote Sens. Environ., 128, 107-117, `https://doi.org/10.1016/j.rse.2012.10.004`, 2013

Kumar, S., and Coauthors: Assimilation of remotely sensed soil moisture and snow depth retrievals for drought estimation, J. Hydrometeor., 15, 2446-2469, `https://doi.org/10.1175/JHM-D-13-0132.1`, 2014.

Kumar, S., C. D. Peters-Lidar, K. R. Arsenault, A. Getirana, D. Mocko, and Y. Liu: Quantifying the added value of snow cover area observations in passive microwave snow depth data assimilation, J. Hydrometeor, 16, 1736-1741, `https://doi.org/10.1175/JHM-D-15-0021.1`, 2015.

Liu, Y., C. D. Peters-Lidard, S. Kumar, J. L. Foster, M. Shaw, Y. Tian, and G. M. Fall: Assimilating satellite-based snow depth and snow cover products for improving snow predictions in Alaska, Adv. Water Resour., 54, 208-227, `https://doi.org/10.1016/j.advwatres.2013.02.005`, 2013.

Nester, T., R. Kirnbauer, J. Parajka, and G. Bloschl: Evaluating the snow component of a flood forecasting model, Hydrol. Res., 43, 762-779, `https://doi.org/10.2166/nh.2012.041`, 2012.

Niu, G.-Y., and Z.-L. Yang: An observation-based formulation of snow cover fraction and its evaluation over large North American river basins, J. Geophys. Res., 112, D21101, `https://doi.org/10.1029/2007JD008674`, 2007.

Parajka, J., and G. Bloschl: The value of MODIS snow cover data in validating and calibrating conceptual hydrologic models, J. Hydrol., 358, 240-258, 2008.

Reichle, R. H., R. D. Koster, G. J. M. Lannoy, B. A. Forman, Q. Liu, and S. P. P. Mahanama: Assessment and enhancement of MERRA land surface hydrology estimates, J. Climate, 24, 6322-6338, `https://doi.org/10.1175/JCLI-D-10-050331.1`, 2011.

Riggs, G. A., D. K. Hall, and M. O. Roman: Overview of NASAs MODIS and visible infrared imaging radiometer suite (VIIRS) snow-cover earth system data records, Earth Syst. Sci. Data, 9, 765-777, `https://doi.org/10.5194/essd-9-765-2017`, 2017.

Rodell, M., and P. R. Houser: Updating a land surface model with MODIS-derived snow cover, J. Hydrometeor., 5, 1064-1075, `https://doi.org/10.1175/JHM-395.1`, 2004.

Toure, A. M., M. Rodell, Z. L. Yang, H. Beaudoing, E. Kim, Y. Zhang, and Y. Kwon: Evaluation of the snow simulations from the Community Land Model, version 4 (CLM4), J. Hydrometeor., 17, 153-170, `https://doi.org/10.1175/JHM-D-14-0165.1`, 2016.

Yang, Z.-L., and Coauthors: The community Noah land surface model with multiparameterization options (Noah-MP): 2. Evaluation over global river basins. J. Geophys. Res., 116, D12100, `https://doi.org/10.1029/2010JD015140`, 2011.

Zhou, X., and Coauthors: Evaluation of Arctic land snow cover characteristics, surface albedo, and temperature during the transition seasons from regional climate model simulations and satellite data, Adv. Meteorol., `https://doi.org/10.1155/2014/604157`, 2014.

---

## Editor Comment (EC1) · Thomas Mölg (Editor) · 10 Apr 2019

Dear colleagues,

Two referees have evaluated the manuscript, and I would like to thank both of them for their thoughtful comments. I would also like to thank the authors for their detailed response and suggestions of how to address the referee comments in a revised MS.

The fact is that both referees express major concerns, and I can understand their main comments and doubts about the present version of the study. In this regard the author responses are not convincing enough in my opinion. I see some remaining, substantial

weaknesses in connection with the following.

(1) Both referees criticized the lack of depth in motivation and in novelty of the study. The authors responded along the lines that several previous studies conducted a similar type of comparison, but that is not a convincing argument. Although I agree entirely with the authors that not every study can/must be embedded in a large international project to feed the motivation part, it would be necessary that some novel or additional aspects are added to a new study (e.g. as suggested in item (2) below). Without doing so, the manuscript will receive little attention.

(2) One of the important goals and motivations will not be met. The authors state that they intend to examine "different physical processes in the LSMs", and I agree that this would be worthwhile and would add flavor to the study that would diminish the weakness expressed in (1) above. However, the whole study is strongly descriptive, and only occasional sentences on qualitative interpretations of what physical processes could be acting in the background are contained. To elevate the value of the present study, the authors should analyze the snow energy budgets and internal snow processes (if resolved in the LSM) by a coordinated quantitative design. Only this would enable to understand how the different snow cover distributions evolve with different LSMs.

(3) The referees also suggest that the meteorological drivers above the snow surface must be evaluated, and I fully agree in particular with regard to item (2) above (it is another important ingredient for "understanding"). The authors are right by saying that not every study can conduct its own measurements and measure all components (e.g., radiation terms). Yet there are several global data sets that cover at least the basic drivers like air temperature and precipitation, based on observations. I would see ERA5 as the right choice if the former would not exist, but the authors should try first to evaluate their results against measurements of atmospheric drivers (not reanalysis).

Since (1)-(3) are substantial, I am sorry to say that I discourage submission of the proposed revised manuscript at this stage. Please note that none of the negative

points raised here are a critique of the basic value of your work, but they are meant to help you improve the details of your study and, therefore, increase the likelihood that your final version of this work will receive more attention and have more impact. In this context, I hope that the reviews and my comments provide you with valid input to work on an improved version of the study, for which you are welcome to consider TC again as a publication platform.

Thomas Mölg

Handling Editor & Co-Editor-In-Chief TC